

# Sky camera geometric calibration using solar observations

**B. Urquhart, B. Kurtz, and J. Kleissl**

Department of Mechanical and Aerospace Engineering, University of California, San Diego, USA

Received: 18 September 2015 – Accepted: 14 October 2015 – Published: 15 January 2016

Correspondence to: J. Kleissl (jkleissl@ucsd.edu)

Published by Copernicus Publications on behalf of the European Geosciences Union.

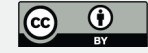

**AMTD**

doi:10.5194/amt-2015-277

**Sky camera geometric calibration using solar observations**

B. Urquhart et al.

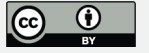

## Abstract

A camera model and associated automated calibration procedure for stationary day-time sky imaging cameras is presented. The specific modeling and calibration needs are motivated by remotely deployed cameras used to forecast solar power production where cameras point skyward and use 180° fisheye lenses. Sun position in the sky and on the image plane provides a simple and automated approach to calibration; special equipment or calibration patterns are not required. Sun position in the sky is modeled using a solar position algorithm (requiring latitude, longitude, altitude and time as inputs). Sun position on the image plane is detected using a simple image processing algorithm. The performance evaluation focuses on the calibration of a camera employing a fisheye lens with an equisolid angle projection, but the camera model is general enough to treat most fixed focal length, central, dioptric camera systems with a photo objective lens. Calibration errors scale with the noise level of the sun position measurement in the image plane, but the calibration is robust across a large range of noise in the sun position. Calibration performance on clear days ranged from 0.94 to 1.24 pixel root mean square error.

## 1 Introduction

The power output variability of renewable energy sources poses challenges to its integration into the electricity grid. Forecasting of renewable power generation (e.g. Monteiro et al., 2009; Perez et al., 2010; Kleissl, 2013) enables more economical and reliable scheduling and dispatch of all generation resources, including renewables, which in turn accommodates a larger amount of variable supply on the electricity grid. Specifically for solar power forecasting, a number of technologies are being applied: numerical weather prediction (e.g. Lorenz et al., 2009; Mathiesen and Kleissl, 2011; Perez et al., 2013); satellite image-based forecasting (e.g. Hammer et al., 1999; Perez and Hoff, 2013); and stochastic learning methods (e.g. Bacher et al., 2009; Marquez and

Coimbra, 2011; Pedro and Coimbra, 2012). For very short term (15 min ahead) solar power forecasting on the kilometer scale, sky imaging from ground stations has demonstrated utility (Chow et al., 2011; Urquhart et al., 2013; Marquez and Coimbra, 2013; Yang et al., 2014).

Some of these sky imaging methods require the camera to be geometrically calibrated, i.e., each pixel must be associated with a corresponding view direction. Together with cloud height estimates, the view direction allows geolocation of clouds and their shadow projections such that their position is known relative to solar power plants. Geometric calibration is a common task in photogrammetry and computer vision, and calibration methods have been developed for a variety of applications. Some methods for calibrating a stationary camera require the use of calibration equipment or setups (Tsai, 1987; Weng et al., 1992; Heikkilä and Silvén 1996; Shah and Aggarwal, 1996) or planar targets (Wei and Ma, 1993; Sturm and Maybank, 1999; Zhang, 2000). Geometric scene information can be used to calibrate the camera's internal parameters (Liebowitz and Zisserman, 1998) or estimate lens distortion (Brown, 1971; Devernay and Faugeras, 2001; Tardif et al., 2006). Scenes with parallel or perpendicular lines or primitive shapes are not generally available for skyward pointing cameras and thus there are no structures from the built environment around which to base a generic and automated calibration procedure.

Cameras used for solar power forecasting often employ fisheye lenses, which require appropriate camera modeling and associated model parameter estimation methods due to the large distortion required to achieve the approximately 180° field of view. Many models which include lens distortion cannot account for distortion present in lenses which have a field of view equal to or exceeding 180° because they rely on converting "distorted" image coordinates (which are finite measurements on the image plane) to "undistorted" image coordinates which are infinite at angles 90° from the optical axis (e.g. Tsai, 1987). Gennery (2006) and Kannala and Brandt (2006) propose generic camera models suitable for fisheye lenses, and the form of the camera model presented here has features of both. The goal of the current work is to develop

**AMTD**

doi:10.5194/amt-2015-277

**Sky camera geometric calibration using solar observations**

B. Urquhart et al.

Title Page

Abstract | Introduction

Conclusions | References

Tables | Figures

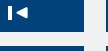 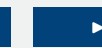

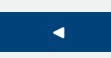 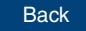

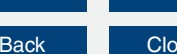

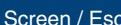

(1) a general camera model for a fixed-focal length wide angle dioptric sky camera with a photo objective lens and (2) a calibration method that can be automated with little user input.

The calibration approach taken here is sometimes referred to as stellar calibration, where the 3-D position of an object or set of objects is treated as known. In particular the sun position in the sky is treated as a known input which is used along with the corresponding measured sun position in an image to calibrate a stationary camera of fixed focal length. Sun position has been used previously for camera calibration. Lalonde et al. (2010) have used manual image annotation to select the sun position in a few images, and with this estimated the focal length, principle point, and two of the three rotational degrees of freedom (the camera horizontal axis was assumed parallel to the ground). The work presented here builds on this idea and extends it using a more generalized camera model and automated sun detection. The camera model here allows any pose, non-square pixels, and both radially symmetric and decentering distortion components.

The layout of this paper is as follows. Section 2 discusses the forward and backward camera model. Section 3 discusses the imaging equipment and solar position input used for the calibration process. Section 4 provides details of the calibration procedure: initialization, linear estimation, and nonlinear estimation. Section 5 provides results for both measured solar position input and synthetic data. Synthetic data is used to assess the uncertainty in calibration performance and parameter estimation as a function of measurement uncertainty.

## 2 Camera model

The forward camera model projects points from a 3-D scene onto the image plane. The backward camera model described in Sect. 2.2. projects points on the image plane to rays in 3-space. Both models are developed assuming that the camera-lens system is

**AMTD**

doi:10.5194/amt-2015-277

**Sky camera geometric calibration using solar observations**

B. Urquhart et al.

central, i.e. all refracted rays within the lens pass through a single point. This, while not physically accurate, yields a close approximation (Ramalingam et al., 2005).

## 2.1 Forward camera model

### 2.1.1 Projective transformation camera model

The standard model for a camera without distortion is a 3-D to 2-D projective transformation, mapping points $X = (X, Y, Z, T)^\top$ in $\mathbb{P}^3$ to $x = (x, y, w)^\top$ in $\mathbb{P}^2$:

$$x = PX, \tag{1}$$

where $P$ is a $3 \times 4$ perspective projection transformation with 11 degree of freedom (it is defined up to scale), and $\mathbb{P}^n$ is the $n$th dimension of projective space. The points $X \in \mathbb{P}^3$ and $x \in \mathbb{P}^2$ are homogeneous quantities and thus are defined only up to scale. The corresponding inhomogeneous points in Euclidean space are $\tilde{X} = (X/T, Y/T, Z/T)^\top = (\tilde{X}, \tilde{Y}, \tilde{Z})^\top$, $\tilde{X} \in \mathbb{R}^3$, and $\tilde{x} = (x/w, y/w)^\top = (\tilde{x}, \tilde{y})^\top$, $\tilde{x} \in \mathbb{R}^2$. The tilde overbar indicates inhomogeneous coordinates throughout this work. When scale factors $T$ or $w$ are zero, the corresponding Euclidean point is infinite. For points not lying on the plane or line at infinity, we can write $X = (\tilde{X}^\top, 1)^\top$ and $x = (\tilde{x}^\top, 1)^\top$, respectively. The point imaging transformation $P$ is given by a composition of Euclidean, affine and perspective transformations

$$P = \underbrace{\begin{bmatrix} \alpha_x & s & x_o \\ 0 & \alpha_y & y_o \\ 0 & 0 & 1 \end{bmatrix}}_{\text{affine}} \underbrace{\begin{bmatrix} 1 & 0 & 0 & 0 \\ 0 & 1 & 0 & 0 \\ 0 & 0 & 1 & 0 \end{bmatrix}}_{\text{perspective}} \underbrace{\begin{bmatrix} r_{11} & r_{12} & r_{13} & t_x \\ r_{21} & r_{22} & r_{23} & t_y \\ r_{31} & r_{32} & r_{33} & t_z \\ 0 & 0 & 0 & 1 \end{bmatrix}}_{\text{Euclidean}}, \tag{2}$$

where the affine transformation is known as the camera calibration matrix (denoted by $K$, parameters defined later), the perspective transformation $P_n = [I|0]$ projects 3-D

**AMTD**

doi:10.5194/amt-2015-277

**Sky camera geometric calibration using solar observations**

B. Urquhart et al.

space points to 2-D image points, and the Euclidean transformation gives the rotation and displacement of the camera center relative to the world coordinate system. The Euclidean transformation can be written in block matrix notation as

$$\begin{bmatrix} \mathbf{R} & t \\ \mathbf{0}^\top & 1 \end{bmatrix},$$

where the upper left block $\mathbf{R}$ with components $r_{ij}$ is a rotation from world coordinates into the camera coordinate system, and the upper right block $t$ with components $t_i$ is a translation giving the displacement from the origin of the camera coordinate system (i.e. the camera center) to the origin of the world coordinate system. The rotation matrix has only three degrees of freedom and can be represented by the angle-axis three vector $w$, where $\mathbf{R} = \mathrm{expm}([w]_\times)$; expm is the matrix exponential and the notation $[\bullet]_\times$ indicates the $3 \times 3$ skew symmetric matrix corresponding to the vector argument. The three rotation plus three translation parameters are known as the camera's extrinsic parameters. In inhomogeneous coordinates, the rigid body (Euclidean) transformation from world coordinates $\tilde{X}$ to camera coordinates $\tilde{X}_{\mathrm{cam}}$ is

$$\tilde{X}_{\mathrm{cam}} = \mathbf{R}\tilde{X} + t,$$

or equivalently using homogeneous coordinates

$$X_{\mathrm{cam}} = \begin{bmatrix} \mathbf{R} & t \\ \mathbf{0}^\top & 1 \end{bmatrix} X. \tag{3}$$

The calibrated points $\hat{x} = (\hat{x}, \hat{y}, \hat{w})^\top$ on the image plane given by $\hat{x} = [\mathbf{I}|\mathbf{0}]\, X_{\mathrm{cam}}$, can be converted to pixel coordinates $x = (x, y, w)^\top$ by the affine transformation $\mathbf{K}$

$$X = \begin{bmatrix} \alpha_x & s & x_o \\ 0 & \alpha_y & y_o \\ 0 & 0 & 1 \end{bmatrix} \hat{x} = \mathbf{K}\hat{x} \tag{4}$$

Discussion Paper | Discussion Paper | Discussion Paper | Discussion Paper | Discussion Paper

**AMTD**

doi:10.5194/amt-2015-277

**Sky camera geometric calibration using solar observations**

B. Urquhart et al.

where $\alpha_x$ and $\alpha_y$ are the effective focal lengths (in pixels) in the $x$ and $y$ directions, respectively, $(x_o, y_o)^\top = \boldsymbol{x}_o$ is the principal point (i.e. the point of intersection of the optical axis with the image plane), and $s$ is the skewness of the pixel coordinate axes. The five parameters in matrix $\mathbf{K}$ are known as the camera's intrinsic parameters. The effective focal lengths $\alpha_x = f k_x$ and $\alpha_y = f k_y \csc\psi$ account for the actual focal length $f$ (in meters) and the potential for pixel sizes $1/k_x$ and $1/k_y$ (in meters per pixel) to vary in the $x$ and $y$ directions, respectively. The angle $\psi$ is the angle between the $x$ and $y$ axes, which is close to $\pi/2$ for our camera, thus $\alpha_y \approx f k_y$. The skewness $s = \alpha_x \cot\psi$ is the degree to which the rows and columns of the image sensor are not orthogonal.

In summary, the model of a camera given by Eq. (1) contains six extrinsic (external) and five intrinsic (internal) camera parameter's and thus has 11 degree of freedom. While the perspective projection camera model has been widely used, it does not account for lens distortion and assumes that the camera is a central projection camera. Since we seek to develop a model for use with a fisheye lens exhibiting a significant amount of distortion, the above model must be modified appropriately.

### 2.1.2 Distortion model

An equivalence class $\boldsymbol{X}_{\mathrm{cam}} \in \mathbb{P}^3$ in projective space (i.e. a point or vector in $\mathbb{P}^3$) defines a ray $\boldsymbol{\Phi} = (\theta, \phi)^\top$ in Euclidean space ($\mathbb{R}^3$):

$$\boldsymbol{\Phi} = \begin{bmatrix} \theta \\ \phi \end{bmatrix} = \begin{bmatrix} \mathrm{atan}\left( \frac{\sqrt{X_{\mathrm{cam}}^2 + Y_{\mathrm{cam}}^2}}{Z_{\mathrm{cam}}} \right) \\ \mathrm{atan}\left( \frac{Y_{\mathrm{cam}}}{X_{\mathrm{cam}}} \right) \end{bmatrix}, \tag{5}$$

where $\theta$ is the angle between the ray and the optical axis (i.e. the camera zenith angle), and $\phi$ is the angle from the positive $X_{\mathrm{cam}}$ axis to the projection of the ray onto the $X_{\mathrm{cam}}$-$Y_{\mathrm{cam}}$ plane. The angle $\phi$ is positive in the counterclockwise direction. The incoming ray

Discussion Paper | Discussion Paper | Discussion Paper | Discussion Paper

**AMTD**

doi:10.5194/amt-2015-277

**Sky camera geometric calibration using solar observations**

B. Urquhart et al.

Discussion Paper | Discussion Paper | Discussion Paper | Discussion Paper |

**AMTD**

doi:10.5194/amt-2015-277

**Sky camera geometric calibration using solar observations**

B. Urquhart et al.

$\Phi$ is mapped onto the image plane by a mapping $\mathcal{D}$ as

$$\begin{bmatrix} \hat{\tilde{x}} \\ \hat{\tilde{y}} \end{bmatrix} = \mathcal{D}(\Phi), \tag{6}$$

where $\left(\hat{\tilde{x}}, \hat{\tilde{y}}\right)^{\top}$ are calibrated inhomogeneous coordinates in the image plane (the hat $^\wedge$ denotes a calibrated point, and the tilde $\sim$ denotes an inhomogeneous coordinate, defined previously). The mapping $\mathcal{D}$ is, in general, nonlinear and includes the distortion produced by the lens-camera system. Here we model $\mathcal{D}$ following Brown (1971) as

$$\mathcal{D}(\Phi) = \hat{r}(\theta) \begin{bmatrix} \cos\phi \\ \sin\phi \end{bmatrix} + \begin{bmatrix} \delta_{cx}(\Phi) \\ \delta_{cy}(\Phi) \end{bmatrix} \tag{7}$$

where $\hat{r}(\theta)$ is the normalized radius on the image plane, and $\delta_{cx}$ and $\delta_{cy}$ account for decentering distortion in the $x$ and $y$ directions, respectively. The normalized radial distance $\hat{r}$ is obtained by dividing the actual radial distance in the image plane by the focal length $f$. These terms will be further discussed in the following subsections.

**Radially symmetric distortion**

The most common form of distortion in dioptric imaging systems with a photo objective lens is radially symmetric distortion. Several adjustments to the perspective projection model to account for radially symmetric distortion have been proposed for small field of view lenses exhibiting moderate amounts of pincushion or barrel distortion (e.g. Slama, 1980). In order to generate a one-to-one mapping of hemispherical radiance (180° field of view) to the image plane, fisheye lenses must introduce extreme radial distortion. For a centered lens system, $\delta_{cx}$ and $\delta_{cy}$ can be taken as zero and $\hat{r}(\theta)$ can be set to one

of the following projection functions (Miyamoto, 1964):

$$g_o(\theta) = \tan(\theta), \qquad \text{perspective projection (not fisheye),} \qquad (8a)$$

$$g_o(\theta) = \theta, \qquad \text{equidistant projection,} \qquad (8b)$$

$$g_o(\theta) = 2\sin(\theta/2), \qquad \text{equisolid angle projection,} \qquad (8c)$$

$$g_o(\theta) = 2\tan(\theta/2), \qquad \text{stereographic projection,} \qquad (8d)$$

$$g_o(\theta) = \sin(\theta), \qquad \text{orthographic projection.} \qquad (8e)$$

Equations (8b) to (8e) correspond to fisheye lens projections. Equation (8a) is the undistorted perspective projection (i.e. same projection model as Eq. 1), but can still be used in the camera model and calibration as described here.

Fisheye lens designers generally strive to meet one of the above projections, but due to manufacturing and assembly tolerances, the standard projections (Eqs. 8b–e) only approximate a particular lens-camera system. In order to model wide angle and fisheye lenses more accurately, a number of models have been proposed (e.g. Kannala et al., 2006, and Shah and Aggarwal, 1996). Here, instead of modeling the radially symmetric distortion using a polynomial in $\theta$ (e.g. Kannala et al., 2006), we follow a suggestion by Gennery (2006) and use one of the standard models $g_o(\theta)$ in Eq. (8), and then fit a polynomial to the residual radial distortion as

$$\hat{r}(\theta) = g_o(\theta) + \sum_2^N k_n \theta^n, \qquad (9)$$

where $\hat{r}(\theta)$ is the normalized radius on the image plane, and the polynomial in $k_n$ models deviations from $g_o(\theta)$. In this work, $N$ was set to nine. In Sect. 4 the coefficients $k_n$ are denoted as a vector $\boldsymbol{k}$, where $\boldsymbol{k} \in \mathbb{R}^8$.

## Decentering distortion

In addition to radially symmetric distortion, lenses exhibit tangential distortion. This deviation from the radial alignment constraint (Tsai, 1987) causes the measured azimuth

**AMTD**

doi:10.5194/amt-2015-277

**Sky camera geometric calibration using solar observations**

B. Urquhart et al.

of a point $\varphi$ to differ from its true azimuth $\phi$. Tangential distortion is due in part to a decentering of lens elements (Conrady, 1919; Brown, 1966). Based on the paraxial optics assumption, Conrady (1919) developed the following radial $\delta_{cr}^C$ and tangential $\delta_{ct}^C$ distortion terms arising from decentering for a point located at $(r, \chi)$ on the image plane:

$$\delta_{cr}^C = 3\mathfrak{p}_1 r^2 \cos(\chi - \chi_1) + \mathfrak{p}_2 r (2 + \cos(2(\chi - \chi_2))) + \mathfrak{p}_2 \tag{10a}$$

$$\delta_{ct}^C = \mathfrak{p}_1 r^2 \sin(\chi - \chi_1) + \mathfrak{p}_2 r \sin(2(\chi - \chi_2)), \tag{10b}$$

where $\mathfrak{p}_1$ and $\mathfrak{p}_2$ are constants that determine the magnitude of each centering defect, $r$ is the radius in the image plane taken from the principal point, $\chi_1$ and $\chi_2$ are reference axes for the distortion effects. The constants are proportional to the lens decentering magnitude $\Delta$ as $\mathfrak{p}_1 \propto \Delta$ and $\mathfrak{p}_2 \propto \Delta^2$. Conrady (1919) did not develop terms of higher than first order in $\Delta$, i.e. only terms containing $\mathfrak{p}_1$ were developed. The terms containing $\mathfrak{p}_2$ (investigated for this work) are the only higher order terms that are not constant or symmetric over the image.

The Brown–Conrady distortion model (Brown, 1971) formulates the radial $\delta_{cr}$ and tangential $\delta_{ct}$ decentering distortion components with reference axis $\phi_o$ to be that of maximum tangential distortion with $\phi$ is positive counter clockwise from the $x$ axis ($\chi$ is positive clockwise):

$$\delta_{cr} = 3\mathcal{P} \sin(\phi - \phi_o),$$

$$\delta_{ct} = \mathcal{P} \cos(\phi - \phi_o),$$

where the terms containing $\mathfrak{p}_2$ have been neglected, and the profile function $\mathcal{P} = \mathfrak{p}_1 r^2$. Because Conrady did not develop terms in $\Delta$ of higher order than one, Brown speculated that $\mathcal{P}$ could be extended as a polynomial in even powers of $r$ (written here as a normalized radial distance):

$$\mathcal{P}(\hat{r}) = \sum_1^M J_m \hat{r}^{2m}.$$

**AMTD**

doi:10.5194/amt-2015-277

**Sky camera geometric calibration using solar observations**

B. Urquhart et al.

Expanding the aberrations due to decentering as developed by Conrady (1919), one finds that the second and third order terms in $\Delta$ produce only lower order terms in $r$ (i.e. $r^1$ and $r^0$). The zeroth order term in $\Delta$ (which is thus present in centered lens systems) produces a shift in the image proportional to $r^3$ and is commonly known as pincushion or barrel distortion. Because decentering effects in most lenses are small ($\mathfrak{p}_1 r^2 \sim 10^{-4}$ pixels for our lense), it is reasonable to neglect $\mathfrak{p}_2$.

The use of the Brown–Conrady decentering distortion model for a fisheye lens should only be considered as an expedient for model fitting, and not as a physical description of optical distortion. Conrady derived the decentering formulae following a paraxial method he devised to analytically obtain the five classical Seidel aberrations (Conrady, 1918). Equation (10) are therefore only valid under the small angle approximation $\sin\phi \approx \phi$, and are thus not valid for the large incidence angles in a fisheye lens. Additionally, there is no physical justification for Brown's extrapolation of $\mathcal{P}$ as an even ordered polynomial in $r$ (recall that Conrady's original model had no higher order terms than $r^3$). The retention in this work of the Brown–Conrady decentering distortion model is for model fitting only.

The radial and tangential distortion can be converted to the corresponding Cartesian components as

$$\begin{bmatrix} \delta_{cx} \\ \delta_{cy} \end{bmatrix} = \begin{bmatrix} \cos\phi & -\sin\phi \\ \sin\phi & \cos\phi \end{bmatrix} \begin{bmatrix} \delta_{cr} \\ \delta_{ct} \end{bmatrix},$$

which upon expanding gives

$$\begin{bmatrix} \delta_{cx} \\ \delta_{cy} \end{bmatrix} = \mathcal{P} \begin{bmatrix} -\left(2\cos^2\phi + 1\right)\sin\phi_o + 2\sin\phi\cos\phi\cos\phi_o \\ -2\sin\phi\cos\phi\sin\phi_o + \left(2\sin^2\phi + 1\right)\cos\phi_o \end{bmatrix}. \tag{11}$$

Discussion Paper | Discussion Paper | Discussion Paper | Discussion Paper

Following Brown by taking

$$p_1 = -J_1 \sin \phi_o,$$
$$p_2 = J_1 \cos \phi_o,$$
$$p_m = \frac{J_{m-1}}{J_1}, \quad m > 2$$

it can be easily shown that

$$\delta_{cx}(\mathbf{\Phi}) = \left[ p_1 \left( 1 + 2\cos^2 \phi \right) + 2 p_2 \sin \phi \cos \phi \right] \left( \hat{r}^2 + p_3 \hat{r}^4 + p_4 \hat{r}^6 + \cdots \right), \tag{12a}$$

$$\delta_{cy}(\mathbf{\Phi}) = \left[ 2 p_1 \sin \phi \cos \phi + p_2 \left( 1 + 2\sin^2 \phi \right) \right] \left( \hat{r}^2 + p_3 \hat{r}^4 + p_4 \hat{r}^6 + \cdots \right). \tag{12b}$$

Here only $p_1$ through $p_4$ are used. In Sect. 4, the coefficients $p_n$ are denoted as a vector $\boldsymbol{p}$, where $\boldsymbol{p} \in \mathbb{R}^4$.

### 2.1.3 Forward camera model overview

Summarizing the results of this section, the forward projection of a 3-D space point to 2-D pixel coordinates consists of the following four steps:

1. Euclidean transformation

$$X_{\text{cam}} = \begin{bmatrix} \mathbf{R} & t \\ \mathbf{0}^{\top} & 1 \end{bmatrix} X$$

2. Cartesian to spherical coordinates

$$\mathbf{\Phi} = \begin{bmatrix} \theta \\ \phi \end{bmatrix} = \begin{bmatrix} \text{atan}\left( \sqrt{X_{\text{cam}}^2 + Y_{\text{cam}}^2} / Z_{\text{cam}} \right) \\ \text{atan}(Y_{\text{cam}} / X_{\text{cam}}) \end{bmatrix}$$

Discussion Paper | Discussion Paper | Discussion Paper | Discussion Paper | Discussion Paper |

**AMTD**

doi:10.5194/amt-2015-277

**Sky camera geometric calibration using solar observations**

B. Urquhart et al.

3. Lens-camera projection with distortion

$$\left[\begin{array}{c} \hat{\hat{x}} \\ \hat{\hat{y}} \end{array}\right] = \left(g_o(\theta) + \sum_2^9 k_n \theta^n\right) \left[\begin{array}{c} \cos\phi \\ \sin\phi \end{array}\right]$$

$$+ \left(\hat{r}^2 + p_3\hat{r}^4 + p_4\hat{r}^6\right) \left[\begin{array}{c} p_1\left(1 + 2\cos^2\phi\right) + 2p_2\sin\phi\cos\phi \\ 2p_1\sin\phi\cos\phi + p_2\left(1 + 2\sin^2\phi\right) \end{array}\right]$$

4. Affine transformation

$$\left[\begin{array}{c} x \\ y \\ 1 \end{array}\right] = \left[\begin{array}{ccc} \alpha_x & s & x_o \\ 0 & \alpha_y & y_o \\ 0 & 0 & 1 \end{array}\right] \left[\begin{array}{c} \hat{\hat{x}} \\ \hat{\hat{y}} \\ 1 \end{array}\right]$$

The full camera model is represented by a nonlinear vector-valued function $\boldsymbol{f}$:

$$\tilde{\boldsymbol{x}} = \boldsymbol{f}(\boldsymbol{X}, \boldsymbol{\beta}), \quad \boldsymbol{\beta} = \left(\alpha_x, \alpha_y, s, \boldsymbol{x}_o^\top, \boldsymbol{w}^\top, \boldsymbol{t}^\top, \boldsymbol{k}^\top, \boldsymbol{p}^\top\right)^\top. \tag{13}$$

## 2.2 Backward projection

In many cases, one is given points $\boldsymbol{x}$ in image coordinates and what is needed is the back projection of those points into world coordinates. This is true for the application of solar forecasting where many quantities derived from images are assigned a space angle $\boldsymbol{\Phi}$ according to their image coordinates. For example, Chow et al. (2011) back project cloud positions detected within an image to a 3-D world plane to generate a mapping of the clouds, and subsequently used this cloud map to ray trace cloud shadows. Note that obtaining the distance from the camera to an object in the scene is not possible from a single image because of the projective nature of the imaging process.

The inversion of mapping $\mathcal{D}$ (Eqs. 6 and 7) is the most difficult part of developing a back projection model from a forward projection model, and Kannala et al. (2006)

Discussion Paper | Discussion Paper | Discussion Paper | Discussion Paper | Discussion Paper |

**AMTD**

doi:10.5194/amt-2015-277

**Sky camera geometric calibration using solar observations**

B. Urquhart et al.

suggest a function inversion approach to this end. An alternative is to formulate a separate back projection model and fit it using synthetic data generated from the forward projection.

After converting to calibrated inhomogeneous image coordinates using $\left(\hat{\hat{x}}, \hat{\hat{y}}, 1\right)^{\top} = \mathbf{K}^{-1}(x, y, 1)^{\top}$, the decentering distortion is formulated as a function of the polar coordinate $(r, \varphi)$ in the image plane

$$\begin{bmatrix} \delta_{cx}(r,\varphi) \\ \delta_{cy}(r,\varphi) \end{bmatrix} = \left(r^2 + q_3 r^4 + q_4 r^6 + q_5 r^8\right) \begin{bmatrix} q_1\left(1 + 2\cos^2\varphi\right) + 2q_2\sin\varphi\cos\varphi \\ 2q_1\sin\varphi\cos\varphi + q_2\left(1 + 2\sin^2\varphi\right) \end{bmatrix},$$
(14)

where

$$\begin{bmatrix} r \\ \varphi \end{bmatrix} = \begin{bmatrix} \sqrt{\hat{\hat{x}}^2 + \hat{\hat{y}}^2} \\ \mathrm{atan}\left(\hat{\hat{y}}/\hat{\hat{x}}\right) \end{bmatrix}.$$
(15)

The residual radially symmetric distortion polynomial (Eq. 9) is reformulated as a function of $r$:

$$\hat{r} - g_o(\theta) = \sum_2^N b_n r^n,$$
(16)

where $\hat{r}$, equivalent to its definition in the forward projection, is the radial coordinate after adjustment for decentering:

$$\hat{r} = \sqrt{\left(\hat{\hat{x}} - \delta_{cx}(r,\varphi)\right)^2 + \left(\hat{\hat{y}} - \delta_{cy}(r,\varphi)\right)^2}$$
(17)

**AMTD**

doi:10.5194/amt-2015-277

**Sky camera geometric calibration using solar observations**

B. Urquhart et al.

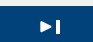

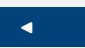 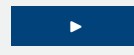

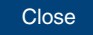

Back | Close

Full Screen / Esc

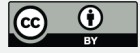

Discussion Paper | Discussion Paper | Discussion Paper | Discussion Paper

$N$ for the back-projection is set to nine. An image point can then be back-projected using

$$\mathbf{\Phi} = \left[ \begin{array}{c} \theta \\ \phi \end{array} \right] = \left[ \begin{array}{c} g_o^{-1}\left( \hat{r} - \sum_2^N b_n r^n \right) \\ \mathrm{atan}\left( \dfrac{\left( \hat{\hat{y}} - \delta_{cy}(r,\varphi) \right)}{\left( \hat{\hat{x}} - \delta_{cx}(r,\varphi) \right)} \right) \end{array} \right]. \tag{18}$$

where inversion of $g_o$ from any of the options listed in Eq. (8) is straightforward. The ray $\mathbf{\Phi}$ can be parameterized in the world reference frame as

$$\left[ \begin{array}{c} X(\lambda) \\ Y(\lambda) \\ Z(\lambda) \\ 1 \end{array} \right] = \left[ \begin{array}{cc} \lambda \mathbf{R}^{\top} & -\mathbf{R}^{\top} \boldsymbol{t} \\ \mathbf{0}^{\top} & 1 \end{array} \right] \left[ \begin{array}{c} \cos\phi\sin\theta \\ \sin\phi\sin\theta \\ \cos\theta \\ 1 \end{array} \right], \tag{19}$$

where $\lambda$ is a scalar. For sky imaging, the camera center is often considered the origin of the world coordinate system and thus $\boldsymbol{t} = \mathbf{0}$.

## 3 Solar position input from sky imager data

### 3.1 Imaging equipment and setup

The University of California, San Diego (UCSD) sky imager (USI) camera system was developed for the purpose of solar power forecasting (Urquhart et al., 2013). The camera is an Allied Vision GE-2040C camera which contains a 15.15 mm × 15.15 mm, 2048 × 2048 pixel Truesense KAI-04022 interline transfer charge coupled device (CCD). The lens is a Sigma circular fisheye lens with a 4.5 mm nominal focal length and equisolid angle projection (Eq. 8c). Images cropped to 1748 × 1748 pixels (3.1 MP) are captured every 30 seconds during daylight hours, which for this experiment yielded over 1400 images per day. The USI uses 3 exposures at integration times of 3, 12, and

Discussion Paper | Discussion Paper | Discussion Paper | Discussion Paper |

**AMTD**

doi:10.5194/amt-2015-277

**Sky camera geometric calibration using solar observations**

B. Urquhart et al.

48 ms to generate a composite HDR image. The system clock is regularly updated using the network time protocol, so image capture times are accurate to within a second. Extensive details of the USI can be found in Urquhart et al. (2015).

The USI used in this work was deployed at the Department of Energy, Atmospheric Radiation Measurement (ARM) Program, Southern Great Plains (SGP) Climate Research Facility from 11 March 2013 to 4 November 2013 at a longitude, latitude, altitude of 97.484856° W, 36.604043° N, 318 m. The horizon around the SGP site is free of mountainous terrain, thus the USI has a nearly 180° field of view of the sky. The camera nominally points straight up, but has a slight angular offset due to the ground not being perfectly level. No leveling of the equipment was performed. Figure 1 shows the USI on its portable mounting stand.

## 3.2 Solar position modeling

The input used in calibrating the camera model (i.e. fitting the camera model parameters) is the angular position of the sun $\Phi_s = (\theta_s, \phi_s)^\top$ and the corresponding position of the sun in a sky image $x_s = (x_s, y_s)^\top$. The angular solar position $\Phi_s$ is estimated using the NREL solar position algorithm (Reda and Andreas, 2004), which adopts the procedure from Meeus (1998). The vector $X_s$ is computed from $\Phi_s$ using Eq. (19) with $\lambda = 1$, $t = 0$, and $R$ is a $3 \times 3$ identity matrix. The NREL algorithm takes observer position (latitude, longitude, altitude) and time as inputs, and outputs the topocentric solar zenith angle $\theta_s$ and topocentric solar azimuth angle $\phi_s$. The refractive index of the air is a function of its density (hence a function of temperature and pressure) along the optical path, and because the atmospheric density gradient is predominantly vertical, the apparent solar zenith angle must be corrected accordingly (Brown, 1964). A correction using annual averages of surface air pressure and temperature is included in the algorithm, and the default value for refraction magnitude at sunrise/sunset is used. The uncertainty on solar zenith angle $\theta_s$ reported by Reda and Andreas is ±0.0003°, and if the image capture time is one second off, the error in solar hour angle would be ±0.004°. In comparison, for our lens a one pixel measurement uncertainty in sun posi-

Discussion Paper | Discussion Paper | Discussion Paper | Discussion Paper

**AMTD**

doi:10.5194/amt-2015-277

**Sky camera geometric calibration using solar observations**

B. Urquhart et al.

**AMTD**

doi:10.5194/amt-2015-277

**Sky camera geometric calibration using solar observations**

B. Urquhart et al.

tion measurements corresponds to approximately $\Delta\theta = 0.14°$ at the horizon. The sun detection process, therefore, introduces significantly more error than the solar position model and time recording errors. For the "full" calibration dataset (case 3, below), 82 % of measurements were within the one pixel measurement uncertainty bounds.

## 3.3 Solar position calibration input

Measurement data consists of automated detection of the sun's position $\boldsymbol{x}_s$ in an image using a set of methods described in Appendix A. The detection process for a single image $i$ results in a set $\{x_s\}_i$ and a set $\{y_s\}_i$ of potential sun coordinates, from each of which the median was taken as the final sun coordinate $\boldsymbol{x}_{s,i} = (x_{s,i}, y_{s,i}, 1)^\top$ to be used for calibration. Here, $i$ is the image index and $\{\bullet\}_i$ represents the set of measurements for image $i$. The detection methods leverage the fact that the sun is the brightest object in a daytime sky image. For the days chosen, the sun could be seen at solar zenith angles near $90°$, indicating that the horizon is at a similar altitude as the instrument. The sun could be detected reliably for images with $\theta_s < 89.625°$. The sun detection algorithm described here was tested on predominantly clear days which simplifies detection because clouds cause occlusion of the sun or saturation of cloudy pixels near the sun.

The sun position is detected in a series of images collected from sunrise to sunset, yielding over 1400 calibration points per day. The set of points collected throughout a single (clear) day nominally forms a smooth arc. To evaluate the camera model and calibration performance under different solar arc input possibilities, five input cases were tested: (1) a single solar arc, (2) two solar arcs on consecutive days, (3) four solar arcs, (4) ten solar arcs with measurement noise due to occasional clouds, (5) a single solar arc with noise due to clouds (Table 1). The solar arcs for cases 1, 4 and 5 are shown in Fig. 2. Case 1 would be preferred in practice as it requires only an – admittedly perfectly clear – day of data. However, limitations in sun position availability during one day may not provide sufficient constraints for the optimization. The improve-

ment associated with adding more days is evaluated in cases 2 and 3. Cases 4 and 5 were designed to provide more realistic and noisy data that would be found in climates without completely clear days.

The sequence of sun position detections forms an arc that should be a smooth curve. The detection process, however, is associated with errors, especially when clouds are present. The deviation of the measured data from a smooth arc can be used to quantify the calibration input error. Separately for each day, a 9th order polynomial is fit to the $x$ and $y$ pixel coordinates as a function of solar hour angle $H$ (obtained from the NREL solar position algorithm). A separate polynomial is obtained for $x$ and $y$, which after obtaining the polynomial coefficients $a_n$ and $b_n$ can be written as

$$\tilde{x}_f = \sum_0^N a_n H^n, \tag{20a}$$

$$\tilde{y}_f = \sum_0^N b_n H^n, \tag{20b}$$

where $(\tilde{x}_f, \tilde{y}_f)$ is the pixel coordinate. A separate polynomial must be computed for each day. The standard deviation of the pixel-by-pixel distance between the measurements and the polynomial fit (Eq. 27c) is given in Table 1 as $SD_m$ and is a useful estimate of the sun position error.

## 4 Calibration procedure

The calibration procedure is a three step process: (1) generate a rough estimate of the intrinsic parameters, (2) estimate the camera pose (rotation and translation) assuming one of the projections in Eq. (8), (3) perform a three stage nonlinear parameter estimation using the Levenberg–Marquhardt algorithm to obtain the final intrinsic and extrinsic parameters. Steps 1 and 2 will be described in Sect. 4.1 and step 3 will be discussed in Sect. 4.2. Calibration results are given in Sect. 5.

Discussion Paper | Discussion Paper | Discussion Paper | Discussion Paper | Discussion Paper

**AMTD**

doi:10.5194/amt-2015-277

**Sky camera geometric calibration using solar observations**

B. Urquhart et al.

## 4.1 Model initialization

In order to apply the Levenberg–Marquhardt (LM) algorithm to estimate the model parameters, the parameter vector $\boldsymbol{\beta} = (\alpha_x, \alpha_y, s, \boldsymbol{x}_o^\top, \boldsymbol{w}^\top, \boldsymbol{t}^\top, \boldsymbol{k}^\top \boldsymbol{p}^\top)^\top$ must be initialized with a reasonably close estimate (see Sect. 2 for the definitions of the components of $\boldsymbol{\beta}$). The simple estimate of intrinsic parameters $(\alpha_x, \alpha_y, s, \boldsymbol{x}_o)$ given in Sect. 4.1.1 needs to only be performed once unless the camera or lens is modified. Pose estimation (Sect. 4.1.2) for the extrinsic parameters $(\boldsymbol{w}^\top, \boldsymbol{t}^\top)$ needs to be performed any time the camera is moved. The distortion parameters $(\boldsymbol{k}^\top, \boldsymbol{p}^\top)$ can be initialized to zero vectors.

### 4.1.1 Intrinsic parameter estimation

In whole sky imagery, the entire sky hemisphere is visible and forms an ellipse on the image plane with eccentricity near unity (e.g. Figs. 2 or 3). This enables a simple automated estimation approach. A Hough circle transform is used to obtain the approximate center $\boldsymbol{x}_{img}$ of and radius $r_{img}$ of this near circular ellipse. The principal point $\boldsymbol{x}_o$ is initialized to $\boldsymbol{x}_{img}$. The $x$ and $y$ focal lengths are assumed to be equal, i.e. $\alpha_x = \alpha_y = \alpha$, and are determined using an unnormalized version of Eq. (9) where $r = \alpha g_o(\theta)$:

$$\alpha = r_{img}/g_o(\theta_{max}), \tag{21}$$

where the radius $r_{img}$ from the Hough circle detection process corresponds to the maximum field of view. For the USI, $\theta_{max}$ is taken to be $\pi/2$ and $g_o$ is given by Eq. (8c), thus $\alpha = r_{img}/\sqrt{2}$. Initially it is assumed that $s$ is zero, i.e. the camera pixel axes are orthogonal. The initial estimate of the camera calibration matrix $\mathbf{K}_o$ is then

$$\mathbf{K}_o = \begin{bmatrix} \alpha & 0 & x_{img} \\ 0 & \alpha & y_{img} \\ 0 & 0 & 1 \end{bmatrix}. \tag{22}$$

**AMTD**

doi:10.5194/amt-2015-277

**Sky camera geometric calibration using solar observations**

B. Urquhart et al.

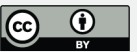

Discussion Paper | Discussion Paper | Discussion Paper | Discussion Paper | Discussion Paper

### 4.1.2 Pose estimation

The camera pose is estimated by computing the linear transformation between the inhomogeneous camera coordinates $\tilde{X}_{\mathrm{cam},i}$ and the homogeneous world coordinates $X_{\mathrm{s},i}$ (from the NREL solar position algorithm, Sect. 3.2):

$$\tilde{X}_{\mathrm{cam},i} = \left[\mathbf{R}|t\right] X_{\mathrm{s},i}, \tag{23}$$

where $i$ is the data point (time/image) index for the set of points to be used in calibration. The camera coordinates $\tilde{X}_{\mathrm{cam},i}$ are obtained by first computing the calibrated image coordinates from the measured sun position: $\hat{x}_i = \mathbf{K}_o^{-1} x_{\mathrm{s},i}$, and then by using

$$\begin{bmatrix} \theta_{\mathrm{cam},i} \\ \varphi_{\mathrm{cam},i} \end{bmatrix} = \begin{bmatrix} g_o^{-1}(\hat{r}_i) \\ \mathrm{atan}\,(\hat{y}_i/\hat{x}_i) \end{bmatrix}, \tag{24}$$

where $\hat{r}_i$ is given by Eq. (17) with decentering distortion set to zero, and finally by projecting onto the unit sphere:

$$\tilde{X}_{\mathrm{cam},i} = \begin{bmatrix} X_{\mathrm{cam},i} \\ Y_{\mathrm{cam},i} \\ Z_{\mathrm{cam},i} \end{bmatrix} = \begin{bmatrix} \sin(\theta_{\mathrm{cam},i})\cos(\varphi_{\mathrm{cam},i}) \\ \sin(\theta_{\mathrm{cam},i})\sin(\varphi_{\mathrm{cam},i}) \\ \cos(\theta_{\mathrm{cam},i}) \end{bmatrix}. \tag{25}$$

The calibrated perspective projection matrix $\hat{\mathbf{P}} = \left[\mathbf{R}|t\right]$ from Eq. (23) can be obtained using the Direct Linear Transform (DLT) algorithm. Premultiplying Eq. (23) by the left nullspace of $\tilde{X}_{\mathrm{cam},i}$:

$$\left[\tilde{X}_{\mathrm{cam},i}^{\top}\right]^{\perp\top} \hat{\mathbf{P}} X_{\mathrm{s},i} = \mathbf{0},$$

where $[\bullet]^{\perp}$ denotes the nullspace of the argument (and thus transposing the input gives the left nullspace). Applying the vec operator yields

$$\left(X_{\mathrm{s},i}^{\top} \otimes \left[\tilde{X}_{\mathrm{cam},i}^{\top}\right]^{\perp\top}\right) \mathrm{vec}\left(\hat{\mathbf{P}}\right) = \mathbf{0},$$

Discussion Paper | Discussion Paper | Discussion Paper | Discussion Paper

AMTD

doi:10.5194/amt-2015-277

Sky camera geometric calibration using solar observations

B. Urquhart et al.

where $\otimes$ is the Kronecker product. Our design matrix then contains subblocks $\mathbf{A}_i = X_{\text{s},i}^\top \otimes \left[\tilde{X}_{\text{cam},i}^\top\right]^{\perp\top}$. Stacking rows $\mathbf{A}_i$ to form a matrix $\mathbf{A}$ gives the homogeneous linear equation

$$\mathbf{A}\hat{p} = \mathbf{0},$$

where $\hat{p} = \text{vec}\left(\hat{\mathbf{P}}\right)$. Due to measurement noise, the right hand side is not identically zero. A least squares solution is obtained by computing the singular value decomposition of $\mathbf{A}$ and taking $\hat{p}$ as the right singular vector corresponding to the smallest singular value (Hartley and Zisserman, 2003). As always with the DLT algorithm, appropriate data normalization is required (Hartley, 1997). $\hat{\mathbf{P}}$ is obtained from $\hat{p}$ by inverting the
vec operation.

Due to imperfect data, the left $3 \times 3$ subblock of $\hat{\mathbf{P}}$ is likely not an orthogonal matrix in $SO(3)$ as is required for rotation matrices. To obtain $\mathbf{R}$ and $t$ from the DLT estimate of $\hat{\mathbf{P}}$ we take $\hat{\mathbf{P}} = \left[\mathbf{M}|\boldsymbol{v}\right]$ where $\mathbf{M}$ is the $3 \times 3$ left subblock of $\hat{\mathbf{P}}$ and $\boldsymbol{v}$ is the rightmost column vector. We then use singular value decomposition to write $\mathbf{M} = \mathbf{U}\mathbf{D}\mathbf{V}^\top$ where $\mathbf{U}$ com-
15 prises the left singular vectors and $\mathbf{V}$ comprises the right singular vectors of $\mathbf{M}$ (both $\mathbf{U}$ and $\mathbf{V}$ are orthogonal matrices), while $\mathbf{D}$ contains the singular values of $\mathbf{M}$. An orthogonal matrix in $SO(3)$ is obtained by taking $\mathbf{R} = \mu\mathbf{U}\mathbf{V}^\top$ where $\mu = \text{sign}\left(\det\left(\mathbf{U}\mathbf{V}^\top\right)\right)$. This gives the closest matrix $\mathbf{R}$ to $\mathbf{M}$ in the sense of the Frobenius norm. The translation vector $t$, which is nominally zero here, is given by $t = -\mathbf{R}\left[\widetilde{\hat{\mathbf{P}}}\right]^\perp$, where the tilde indicates
that after computing the nullspace, the resulting homogeneous 4-vector is converted to an inhomogeneous 3-vector before multiplication by $\mathbf{R}$.

## 4.2 Nonlinear optimization of model parameters

Using Eq. (13) we define an error function $\boldsymbol{\epsilon}_i\left(\boldsymbol{\beta}\right)$ for a single measurement:

$$\boldsymbol{\epsilon}_i\left(\boldsymbol{\beta}\right) = f\left(X_{\text{s},i}, \boldsymbol{\beta}\right) - \tilde{x}_{\text{s},i},$$

**AMTD**

doi:10.5194/amt-2015-277

**Sky camera geometric calibration using solar observations**

B. Urquhart et al.



Discussion Paper | Discussion Paper | Discussion Paper | Discussion Paper | Discussion Paper |

where $f$ is a function that projects world coordinates $X_{s,i}$ to image coordinates $\tilde{x}_i = f\left(X_{s,i}, \beta\right)$ as a function of parameters $\beta$. What we seek is a parameter vector $\beta$ such that $\|\boldsymbol{e}(\beta)\|^2$ is minimum, where $\boldsymbol{e} = \left(\boldsymbol{e}_1^\top \ldots, \boldsymbol{e}_M^\top, \boldsymbol{e}_c\right)^\top$. The vector $\boldsymbol{e}_c(\beta)$ is a penalty vector defined in Sect. 4.2.1, and $M$ is the number of measurements. The model is fit by minimizing the sum of squared distances between the measured and modeled inhomogeneous pixel coordinates:

$$\|\boldsymbol{e}(\beta)\|^2 = \sum_i^M d\left(\tilde{x}_i, \tilde{x}_{s,i}\right)^2 + \boldsymbol{e}_c^\top \boldsymbol{e}_c = \sum_i^M \left\|\tilde{x}_i - \tilde{x}_{s,i}\right\|^2 + \boldsymbol{e}_c^\top \boldsymbol{e}_c, \tag{26}$$

where $d$ is the Euclidean distance function. In the case of the synthetic data of Sect. 5.3, the sum of squared distances is taken for the ground truth data with noise added $\tilde{\bar{x}}_i + \tilde{\delta}_i$ (representing noised measurements) and the modeled data $\tilde{x}_i$ (i.e. $\sum_i^N d\left(\tilde{x}_i, \tilde{\bar{x}}_i + \tilde{\delta}_i\right)^2$). The nonlinear calibration of the forward model is accomplished by using the Levenberg–Marquhardt (LM) algorithm, for which an excellent introduction is given in Hartley and Zisserman (2003).

The calibration is performed in three successive stages: (1) take $k = 0$ and $p = 0$, i.e. do not include residual radial distortion and decentering distortion, (2) include residual radial distortion terms $k$, but take $p = 0$, (3) include both residual radial distortion $k$ and decentering distortion $p$. Three stages of nonlinear optimization were used because it was found that this approach was more consistent across the different test cases. The multi-stage optimization process first fits the "basic" model parameters $(\alpha_x, \alpha_y, s, x_o^\top, w^\top, t^\top)$ and does not include corrections to the standard distortion model $g_o(\theta)$. Additional degrees of complexity are sequentially added (i.e. radial (stage 2) followed by decentering distortion (stage 3)). The motivation in doing so is to avoid local minima that would result in errors in the estimation of the basic parameters. Intrinsic and extrinsic parameter estimates for stage 1 initialization are given in Sects. 4.1.1 and 4.1.2, and the subsequent stages are initialized with the results of the previous stage.

Discussion Paper | Discussion Paper | Discussion Paper | Discussion Paper | Discussion Paper |

**AMTD**

doi:10.5194/amt-2015-277

**Sky camera geometric calibration using solar observations**

B. Urquhart et al.

Title Page

Abstract | Introduction

Conclusions | References

Tables | Figures

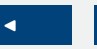 | 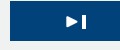

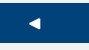 | 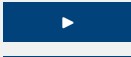

### 4.2.1 Calibration constraints

It was found necessary to enforce additional constraints in the model fitting process to ensure consistent and physically significant results. The LM algorithm is a type of unconstrained optimization, so enforcement of constraints is implemented using a penalty vector $\boldsymbol{e}_c(\boldsymbol{\beta}) = (c_1, \ldots, c_n)^\top$ appended to error vector $\boldsymbol{e}$. Constraints are recomputed each iteration of the LM algorithm along with updates to $\boldsymbol{\beta}$.

To ensure that the residual and nominal radially symmetric distortion are orthogonal functions over the field of view, the following constraint was used:

$$c_1 = \int_0^{\theta_{max}} g_o(\theta) \sum_2^N k_n \theta^n \mathrm{d}\theta = \sum_2^N k_n \int_0^{\theta_{max}} g_o(\theta) \theta^n \mathrm{d}\theta,$$

where $c_1 = 0$ indicates orthogonality. Without this constraint, the LM algorithm tended to decrease $\alpha_x$ and $\alpha_y$ and increase the $k_n$ to compensate, leaving the focal lengths at values that were obviously incorrect based on the nominal lens and sensor specifications. This constraint is very important if the formulation in Eq. (9) is to be used for the radially symmetric distortion. The specific shape of the solar arc used to calibrate the camera, particularly when only a single day was used, resulted in a falsely large skewness $s$. This was corrected by applying a penalty $c_2$ on deviations from circularity of the ellipse formed at $\theta_{max}$ parameterized by varying the azimuth angle. A simple metric such as the standard deviation of the radius of the ellipse at different azimuth angles, taken from the center $\boldsymbol{x}_o^\top$ is simple and effective for this purpose. A similar and simpler approach would be to place a penalty $c_2$ that is proportional to $|s|$, however this was not tested in this work. The last constraint applied was that $\hat{r}(\theta)$ was forced to be monotonically increasing with $\theta$ (the lens mapping would not be one-to-one if it was not!) by applying a penalty $c_3$ if $\mathrm{d}\hat{r}(\theta)/\mathrm{d}\theta < 0$.

**AMTD**

doi:10.5194/amt-2015-277

**Sky camera geometric calibration using solar observations**

B. Urquhart et al.

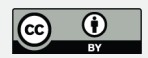

Discussion Paper | Discussion Paper | Discussion Paper | Discussion Paper | Discussion Paper |

**AMTD**

doi:10.5194/amt-2015-277

**Sky camera geometric calibration using solar observations**

B. Urquhart et al.

# 5  Calibration results

## 5.1  Calibration performance metrics

The root mean square error (RMSE), mean absolute error (MAE) and standard deviation (SD) are computed as

$$\text{RMSE} = \left[\frac{1}{M}\sum_{i=1}^{M} d\boldsymbol{x}_i^{\mathsf{T}} d\boldsymbol{x}_i\right]^{1/2} \tag{27a}$$

$$\text{MAE} = \frac{1}{M}\sum_{i=1}^{M} |d\boldsymbol{x}_i|, \tag{27b}$$

$$\text{SD} = \left[\frac{1}{M}\sum_{i=1}^{M}\left(d\boldsymbol{x}_i - \overline{d\boldsymbol{x}_i}\right)^2\right]^{1/2} \tag{27c}$$

where the total number of measurements is $M$, and $d\boldsymbol{x}_i = \tilde{\boldsymbol{x}}_i - \tilde{\boldsymbol{x}}_{\text{s},i}$ is the distance vector from the modeled point $\boldsymbol{x}_i$ to the measured solar position $\boldsymbol{x}_{\text{s},i}$. The vertical bars $|\bullet|$ denote the 2-norm of the argument, and $\overline{d\boldsymbol{x}_i}$ is the mean distance vector for all points $i$. These definitions hold for the evaluation of measurement error as well, where instead $d\boldsymbol{x}_i = \tilde{\boldsymbol{x}}_{\text{s},i} - \tilde{\boldsymbol{x}}_{f,i}$ (see Sect. 3.3 for description of the polynomial fit $\tilde{\boldsymbol{x}}_{f,i}$).

## 5.2  Calibration using the solar position

The results of calibrating the USI for the five different solar arc cases is shown in Table 2. In the cases using more than one solar arc (cases 2–4), the principle point is consistent to within 0.60 pixels (4.4 µm). The $x$ and $y$ focal lengths are consistent to within 0.77 pixels (5.7 µm) for all cases and consistent to within 0.29 pixels (2.1 µm) for cases 2–4. The camera pose results presented here are represented by three angles in Table 2: $\phi_{xz}$ is the angle of rotation of the camera $X_{\text{cam}}$ axis from the world $X$ axis about the world $Z$ axis (effectively the instrument's rotation from a northern alignment); $\theta_{zz}$ is the angle between the camera $Z_{\text{cam}}$ and world $Z$ axis (i.e. the degree to which the

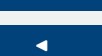
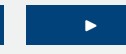
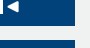
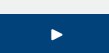
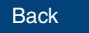


**AMTD**

doi:10.5194/amt-2015-277

**Sky camera geometric calibration using solar observations**

B. Urquhart et al.

system is tilted); and $\psi_z$ is the azimuthal direction towards which the $Z_{cam}$ axis is tilted. The pose determined in all cases was very consistent, with a maximum difference in $\theta_{zz}$ and $\phi_{xz}$ of 0.2°. Because the tilt angle $\theta_{zz}$ was very small, there was increased variability in the tilt direction $\psi_z$ which is to be expected. Anecdotal observations of the USI 1.8 system as deployed at the SGP site indicate that it was tilted slightly southwest and rotated by about 45° with respect to north, which is supported by the estimated pose.

The performance of camera calibration using solar position is given in Table 3 along with the estimated measurement error of the sun position. Calibration error for 1 or more clear days was around 1 pixel (0.93 to 1.24 pixels). Including cloudy days increased the error to 2.9 and 6.3 pixels for the two cloudy cases tested. Including measurement data with more dispersion, as is the case with cloudy days in this study, will always increase the calibration error. This is because for a given set of model parameters, the projection of sun position will form a smooth arc, while the measured sun position will have some dispersion around this arc. Larger dispersion will yield larger calibration error values, which is why it is important to consider calibration error in the context of measurement error.

While not a true lower bound on calibration accuracy, the measurement errors given here can be used to assess the calibration accuracy relative to the estimated accuracy of the input data. The polynomial fit to the measurement data (Eq. 20) does not have the same constraints as fitting the camera model parameters to the measurement data, thus the measurement standard deviation ($SD_m$) and measurement root mean square difference ($RMSD_m$) are lower than the SD and RMSE obtained for camera calibration, with $(RMSE - RMSD_m)/RMSD_m$ lying between 1.8 and 24 %. The proportionality of calibration and measurement error indicated in Table 3, along with the consistency of the parameter estimation (Table 2) indicates that the camera model presented here reasonably approximates the imaging process for the camera and lens tested. It also indicates that the calibration procedure is consistently obtaining reasonable parameter

**AMTD**

doi:10.5194/amt-2015-277

**Sky camera geometric calibration using solar observations**

B. Urquhart et al.

estimates for the model used. Additionally, the robustness of the model and calibration to larger measurement errors and outliers is demonstrated in case 5.

## 5.3 Camera model parameter uncertainty

As with any image detection algorithm, there are errors in the position of the sun obtained from the detection algorithm (Tables 1 and 3). Depending on the content of each image, such as the possibility of thin clouds veiling a still visible sun, or more opaque clouds passing near or occluding the sun, the magnitude of the detection error will vary. A Monte Carlo method was used to assess the uncertainty in model performance and parameter estimation as a function of measurement error. A ground truth synthetic calibration dataset was constructed with $Q = 1673$ data points by simulating a single solar arc on 13 May 2013 (Fig. 3). The points in the world coordinate system $\left\{ \boldsymbol{X}_i = (X_i, Y_i, Z_i, 1)^\top \in \mathbb{P}^3, i = 1\ldots Q \right\}$ were obtained by computing solar position $\boldsymbol{\Phi}_{s,i}$ every 30 seconds from sunrise to sunset with the NREL solar position algorithm (Reda and Andreas, 2004), and then by projecting the solar position onto the unit sphere centered at the camera center. The ground truth pixel coordinates $\left\{ \overline{\boldsymbol{x}}_i = \left( \overline{x}_i, \overline{y}_i, 1 \right)^\top \in \mathbb{P}^2, i = 1\ldots Q \right\}$ were obtained by applying the forward camera model (Sect. 2.1) to points $\boldsymbol{X}_i$. The ground truth camera model parameters were set according to the results from solar calibration case 3. The points $\boldsymbol{X}_i$ were treated as known points in space corresponding to synthetic measurements $\overline{\boldsymbol{x}}_i + \boldsymbol{\delta}_i$, where $\boldsymbol{\delta}_i$ is the measurement error generated as follows. The points $\overline{\boldsymbol{x}}_i$ were taken as the mean measurement values for $Q \times S$ independent normal probability distributions $P_{ij}\left( \overline{\boldsymbol{x}}_i, \sigma_j \right)$ with standard deviations $\sigma_j$ where $j = 1\ldots S$. Standard deviation $\sigma_j$ was varied from 0 to 10 pixels in steps of 0.25 pixels (thus $S = 41$). The synthetic sun measurements $\left( \overline{\boldsymbol{x}}_i + \boldsymbol{\delta}_i \right)_{kj}$ used in calibration trial $kj$ were obtained by sampling $P_{ij}\left( \overline{\boldsymbol{x}}_i, \sigma_j \right)$. A number of trials $N_t = 1000$ was performed for each $j$, yielding $N_t \times S$ calibration trials for each set of $Q$ points. For the $k$th trial at error level $\sigma_j$, a set of model parameters

Discussion Paper | Discussion Paper | Discussion Paper | Discussion Paper | Discussion Paper |

$$\left\{ \boldsymbol{\beta}_{kj} = \left( \alpha_x, \alpha_y, s, \boldsymbol{x}_o^\top, \boldsymbol{w}^\top, \boldsymbol{t}^\top, \boldsymbol{k}^\top, \boldsymbol{p}^\top \right)_{kj}^\top, k = 1 \ldots N_t, j = 1 \ldots S \right\}$$ is obtained. The distribution of $\boldsymbol{\beta}_{kj}$ at each $j$ (i.e. along dimension $k$) is a measure of the uncertainty in the model parameters at error level $\sigma_j$.

The distribution of true root mean square calibration error (RMSE) for $\sigma_j \in [0, 10]$ pixels is shown in Fig. 4. RMSE is computed as

$$\mathrm{RMSE}_{kj} = \left[ \frac{1}{Q} \sum_{i=1}^{Q} \left( \tilde{x}_{i,kj} - \tilde{\bar{x}}_i \right)^2 \right]^{1/2}, \tag{28}$$

where $\tilde{\bar{x}}_i$ is the ground truth pixel position of the $i$th point, and $\tilde{x}_{i,kj}$ is the modeled pixel position of the $i$th point (i.e. the projected pixel position of $\boldsymbol{X}_i$) for calibration trial $k$ at error level $\sigma_j$. Even for $\sigma_j = 10$ pixels, the median RMSE is below 1 pixel. For reference, the measurement error SD in Table 1 for clear days is less than 0.75 pixels, and the worst case tested here (case 5) has a measurement SD of 5.21 pixels. Based on Fig. 4, these measurement errors correspond to true calibration errors of $0.14 \pm 0.03$ pixels and $0.37 \pm 0.11$ pixels (mean $\pm 90\,\%$ confidence interval), respectively, which is considerably lower than the RMSE reported in the first column of Table 3. This assumes measurement errors are normally distributed.

Distributions of parameter estimation for four of the intrinsic parameters are shown in Fig. 5. For both $\alpha_x$ and $\alpha_y$ the 90 % uncertainty bounds are nearly linear. For our camera, this approximately follows $\alpha_{P90} = \overline{\alpha} \pm 0.05 \mathrm{SD}_\mathrm{m}$, which is about 0.09 % error at $\mathrm{SD}_\mathrm{m} = 10$ pixels. The overbar indicates the mean value. Similar results hold for the 90 % uncertainty bounds of $(x_o, y_o)$, which follow $\boldsymbol{x}_{o,P90} = \overline{\boldsymbol{x}}_o \pm 0.25 \mathrm{SD}_\mathrm{m}$ and gives 0.29 % error at $\mathrm{SD}_\mathrm{m} = 10$ pixels. The latter error percentage is computed using $0.25 \mathrm{STD}_\mathrm{m}/N_p \times 100\,\%$, where $N_p = 874$ pixels is the radius of the usable sky image circle (Fig. 3). For the application of solar forecasting using sky imagery, these error levels are satisfactory (at present).

Discussion Paper | Discussion Paper | Discussion Paper | Discussion Paper | Discussion Paper

**AMTD**

doi:10.5194/amt-2015-277

**Sky camera geometric calibration using solar observations**

B. Urquhart et al.

**AMTD**

doi:10.5194/amt-2015-277

**Sky camera geometric calibration using solar observations**

B. Urquhart et al.

# 6  Conclusions

The increasing use of stationary daytime sky imagery instruments for solar forecasting applications has motivated the need to develop automatic geometric camera calibration methods and an associated general camera model. The camera model presented is not specific to fisheye lenses, and is generally applicable to most fixed focal length dioptric camera systems with a photo objective lens. We have proposed a method to automatically detect and use the sun position over a sequence of images to calibrate the proposed camera model. Calibration performance on clear days ranged from 0.94 to 1.24 pixel root mean square error (RMSE). An uncertainty analysis indicated that if measurement errors are normally distributed, this corresponds to a true calibration error of $0.14 \pm 0.03$ to $0.16 \pm 0.03$ pixels RMSE ($0.07 \pm 0.02$ to $0.08 \pm 0.02$ pixels SD), respectively. A back-projection model, which may be more useful for many applications, is proposed as a straightforward extension of the forward projection model. The uncertainty in the forward model parameters was analyzed and is provided graphically as a function of solar position measurement error.

# Appendix A:  Sun position detection

The sun is only detected for images with solar zenith angles $\theta_s < 89.625°$. For approximately $\theta_s < 87°$, the pixels surrounding the sun's location saturate for the USI camera. For the purposes of image detection, this saturated region, which is larger than the sun itself, will be referred to as the "sun" when discussing the image of the daytime sky. The USI uses 3 exposures at different integration times to generate a composite HDR image which reduces the number of saturated pixels encompassing the sun. When the sky is clear, the sun is the only saturated object in the sky which simplifies its detection. With the lens used on the USI, the sun appears as a nearly circular ellipse. The high intensity and near circularity of the sun along with the vertical smear stripe (occurring

in columns containing the sun) are the primary image features used in the sun position detection process.

Both the red-green-blue (RGB) and hue-saturation-value (HSV) color spaces were used for detection, and each color image matrix will be referred to as an X-image, e.g. the R-image (the red image). The approximate diameter of the sun Ø is detected by constructing a binary image by thresholding the V-image at the 99.99th percentile (applicable for our 3.1 MP camera), and then performing an erosion and dilation to remove noise. The diameter of the largest connected binary entity is taken. The apparent sun diameter changes with solar zenith angle, and this size metric is used in constructing detection filters. Three filters are then constructed and subsequently convolved with the V-image: (1) a binary circular kernel of diameter Ø, (2) a Gaussian kernel; and (3) a modified Gaussian kernel which has a flattened top. The standard deviation $\sigma$ (in units of pixels) used for constructing the Gaussian kernels is

$$\sigma = \min\left(3\ln(91° - \theta_s)^2 + 1, 24\right),$$

which was obtained empirically for our camera based on the observed size of the saturated sun area. Kernel 3 was "flattened" such that the circular flat top of the Gaussian was the diameter of the sun Ø. For each kernel, the row $y_s$ and column $x_s$ of the maximum value of the convolution image was taken to be the solar position.

The columns containing the vertical smear (Fig. 2) are detected by extracting the first row of the V-image and the sum of the first row for the R,G and B-images (i.e. three times the first row of the equal weight grayscale image). A measure of the local mean is subtracted from each row separately using a 100 pixel moving average filter. The product of these two rows (pixel-by-pixel product) gives a very strong peak at the smear column which is taken as the column of the sun $x_s$. A sub image extracted from the original image consisting of the set of columns surrounding the sun column ($\sim 10\sigma$ columns) is used for further sun position detection. A Förstner circle detector (Förstner and Gülch 1987) is applied to this sub-image with a window size of $7.5\sigma$ columns and the resulting maximum minor eigenvalue is taken as the sun location.

Discussion Paper | Discussion Paper | Discussion Paper | Discussion Paper | Discussion Paper

**AMTD**

doi:10.5194/amt-2015-277

**Sky camera geometric calibration using solar observations**

B. Urquhart et al.

The detection processes described yield four row-column pairs (three from the circular kernel convolutions and a fourth from the Förstner operator), and the detection of maximum smear gives a fifth column estimate for a total of 4 detected rows $\{y_s\}$ and 5 detected columns $\{x_s\}$ (the braces $\{\bullet\}$ denotes a set of measurements). The median row and column position is taken as the best estimate position. Generally these methods are consistent to within 3 pixels. The detection process would be simpler and more accurate if the camera had been set up to take very short (microsecond) exposures in between regular image capture operations. This would yield a calibration dataset where the saturated sun region would be only a few pixels in diameter instead of 10 s of pixels. This was not available for this work, but is strongly recommended for operational autocalibration when using the solar position as calibration input.

It should be noted that the sun detection method is purely empirical and was not designed to have the fastest performance. In practice, any reasonable algorithm can be used for the sun position detection. If the position errors are zero mean and normally distributed, then the uncertainty analysis in Sect. 5.3 can be used as a guide for expectations of calibration accuracy. The detection method described here is one of many that can be used, and the authors expect that other superior algorithms could be constructed. Since small calibration errors were obtained, the present algorithm is sufficient to demonstrate the calibration methodology.

## Appendix B: Calibration of the backward model

The calibration of backwards projection model parameters was performed with a single stage. The parameter vector used in the LM algorithm consisted only of $b_n$ and $q_n$. The constraints were found to be unnecessary because the process involves fitting only the residual radially symmetric and decentering distortion. The focal lengths $\alpha_x$ and $\alpha_y$ are already set, thus the orthogonality constraint is not required. The other two constraints treat the specific shape of the solar arc, and the back projection parameters are fit using synthetic data points generated from the forward projection which cover

**AMTD**

doi:10.5194/amt-2015-277

**Sky camera geometric calibration using solar observations**

B. Urquhart et al.

the whole image, and therefore are not required. To initialize LM for back-projection fitting, coefficients $b_n$ can be set to the $k_n$ obtained in the forward projection. It was found empirically that $k_n$ are very close to $b_n$ for the equisolid angle lens used on the USI, and should be even closer if an equidistant lens is used. Coefficients $q_n$ can be
set to zero.

*Acknowledgements.* We would like to acknowledge the hardware development team for the USI which consisted of over 25 student volunteers (see Urquhart et al. (2015) for names). In particular we would like to thank Elliot Dahlin for organizing the development team and Mohamed Ghonima for assisting in the deployment of the USI at the Atmospheric Radiation
Measurement (ARM) Program site. We appreciate the funding provided by the US Department of Energy ARM Program to the Southern Great Plains (SGP) site to support this experiment. We are very grateful to the SGP site staff for providing consistent and prompt support throughout our campaign: Rod Soper, John Schatz, Ken Teske, Chris Martin, Tim Grove, David Swank, Pat Dowell. We thank the UCSD von Liebig Center, the California Energy Commission, and the
California Public Utilities Commission California Solar Initiative for providing the authors funding during the preparation of this article.

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

**Table 1.** Calibration test cases. The days included in each test case are given along with the number of sun position points and an estimate of measurement standard deviation $SD_m$ which represents the extent to which the data deviate from a smooth arc. See also Fig. 2 and Eq. (20).

| | Day(s) | points | $SD_m$ [pixels] |
|---|---|---|---|
| case 1 | 13 May | 1582 | 0.4346 |
| case 2 | 13, 14 May | 3195 | 0.4313 |
| case 3 | 13, 14 May | 6543 | 0.6237 |
| | 2, 11 Jun | | |
| case 4 | 13, 14, 22 May | 15 966 | 2.4883 |
| | 1, 2, 3, 7, 9, 10, 11 Jun | | |
| case 5 | 7 Jun | 1482 | 5.2107 |

**AMTD**

doi:10.5194/amt-2015-277

**Sky camera geometric calibration using solar observations**

B. Urquhart et al.

**Table 2.** Camera model parameters (excluding distortion terms) determined from the five test cases of solar input data. The mean and standard deviation are also given. The units denoted [pixels $f^{-1}$] is pixels per focal length ($f$ has units of meters).

| | $\alpha_x$ [pixels $f^{-1}$] | $\alpha_y$ [pixels $f^{-1}$] | $s$ [pixels $f^{-1}$] | $x_o$ [pixels] | $y_o$ [pixels] | $\phi_{xz}$ [deg.] | $\theta_{zz}$ [deg.] | $\psi_z$ [deg.] |
|---|---|---|---|---|---|---|---|---|
| case 1 | 601.44 | 601.44 | $-1.94 \times 10^{-3}$ | 873.24 | 881.62 | 47.89 | 2.58 | 136.51 |
| case 2 | 601.40 | 601.40 | $1.72 \times 10^{-3}$ | 871.48 | 882.97 | 47.90 | 2.47 | 141.48 |
| case 3 | 601.32 | 601.32 | $2.57 \times 10^{-3}$ | 871.96 | 883.00 | 47.91 | 2.46 | 140.16 |
| case 4 | 601.12 | 601.11 | $4.14 \times 10^{-3}$ | 871.97 | 883.31 | 47.93 | 2.43 | 140.06 |
| case 5 | 601.88 | 601.88 | $2.53 \times 10^{-3}$ | 874.20 | 880.84 | 47.90 | 2.63 | 134.02 |
| mean | 601.43 | 601.43 | $1.80 \times 10^{-3}$ | 872.57 | 882.35 | 47.91 | 2.51 | 138.45 |
| SD | 0.25 | 0.25 | $2.03 \times 10^{-3}$ | 1.00 | 0.95 | 0.01 | 0.06 | 2.76 |

**AMTD**

doi:10.5194/amt-2015-277

**Sky camera geometric calibration using solar observations**

B. Urquhart et al.

**Table 3.** Calibration error metrics for each case: root mean square error (RMSE); mean absolute error (MAE); standard deviation (SD); and measurement root mean square difference ($RMSD_m$) and measurement standard deviation ($SD_m$).

| | RMSE [pixels] | [µm] | MAE [pixels] | [µm] | SD [pixels] | [µm] | $RMSD_m$ [pixels] | [µm] | $SD_m$ [pixels] | [µm] |
|---|---|---|---|---|---|---|---|---|---|---|
| case 1 | 0.9370 | 6.931 | 0.7775 | 5.752 | 0.5229 | 3.868 | 0.8099 | 5.991 | 0.4346 | 3.215 |
| case 2 | 0.9635 | 7.128 | 0.8089 | 5.984 | 0.5235 | 3.873 | 0.7802 | 5.772 | 0.4313 | 3.191 |
| case 3 | 1.2381 | 9.159 | 1.0241 | 7.576 | 0.6956 | 5.146 | 1.0538 | 7.795 | 0.6237 | 4.614 |
| case 4 | 2.9351 | 21.712 | 1.4852 | 10.987 | 2.5316 | 18.727 | 2.8080 | 20.772 | 2.4883 | 18.407 |
| case 5 | 6.2994 | 46.510 | 3.4777 | 25.726 | 5.2525 | 38.855 | 6.1871 | 45.769 | 5.2107 | 38.546 |

**AMTD**

doi:10.5194/amt-2015-277

**Sky camera geometric calibration using solar observations**

B. Urquhart et al.

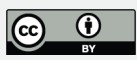

Discussion Paper | Discussion Paper | Discussion Paper | Discussion Paper

**Figure 1.** USI 1.8 in the instrument field at the Department of Energy, Atmospheric Radiation Measurement Program, Southern Great Plains Climate Research Facility.

Discussion Paper | Discussion Paper | Discussion Paper | Discussion Paper

**AMTD**

doi:10.5194/amt-2015-277

**Sky camera geometric calibration using solar observations**

B. Urquhart et al.

Full Screen / Esc

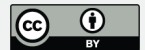

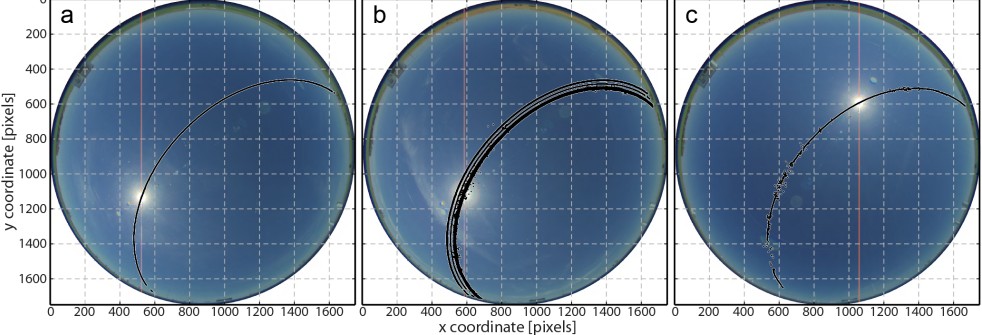

**Figure 2.** Solar position measurements on **(a)** 13 May 2013 (case 1); **(b)** three days in May and seven days in June 2013 with an image on 11 June 2013 (case 4); **(c)** 7 June 2013 (case 5). Measurements are overlaid on example images.

Discussion Paper | Discussion Paper | Discussion Paper | Discussion Paper

**AMTD**

doi:10.5194/amt-2015-277

**Sky camera geometric calibration using solar observations**

B. Urquhart et al.

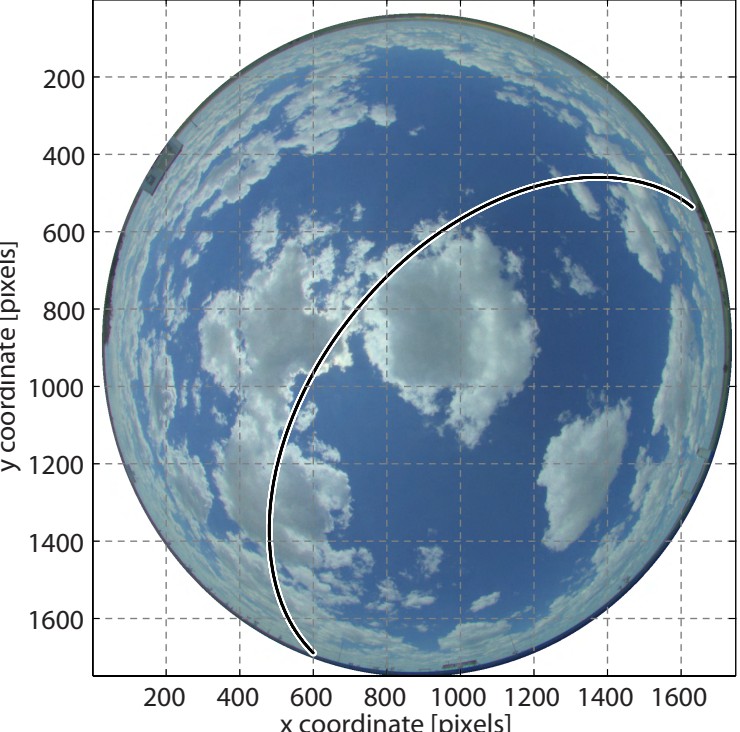

**Figure 3.** Synthetic dataset point distribution. The 1673 points are generated from taking the solar position every 30 seconds from sunrise to sunset on 13 May 2013, and projecting onto the image plane using a set of ground truth camera model parameters. The points shown are ground truth with no noise added. Background image (for visual reference only) is from 3 May 2013.

Discussion Paper | Discussion Paper | Discussion Paper | Discussion Paper |

**AMTD**

doi:10.5194/amt-2015-277

**Sky camera geometric calibration using solar observations**

B. Urquhart et al.

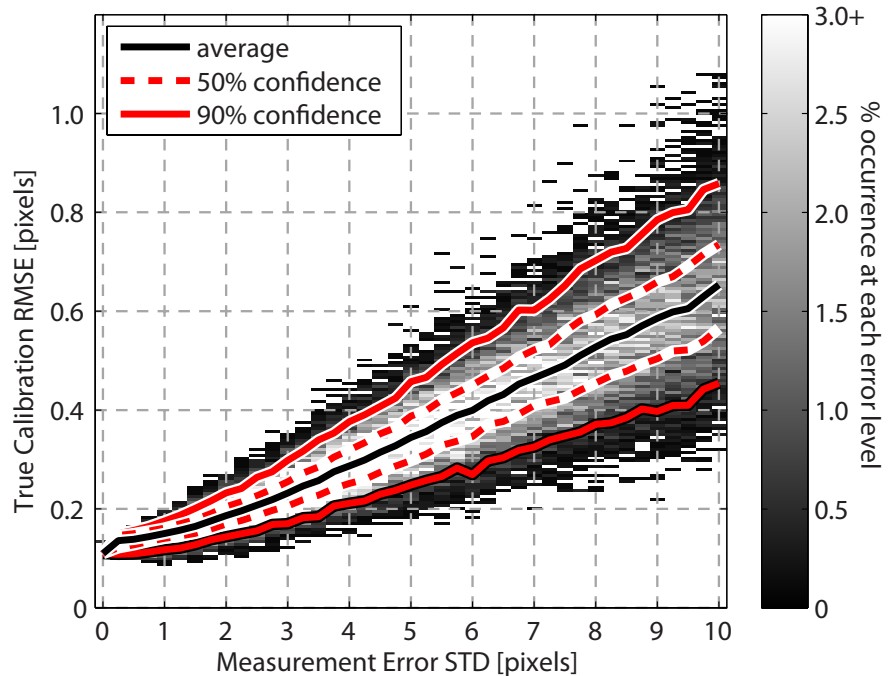

**Figure 4.** Root mean square calibration error distribution (Eq. 27a) as a function of simulated measurement error standard deviation $\sigma_j$. The mean, 50 and 90 % confidence intervals are shown as curves.

Discussion Paper | Discussion Paper | Discussion Paper | Discussion Paper

**AMTD**

doi:10.5194/amt-2015-277

**Sky camera geometric calibration using solar observations**

B. Urquhart et al.

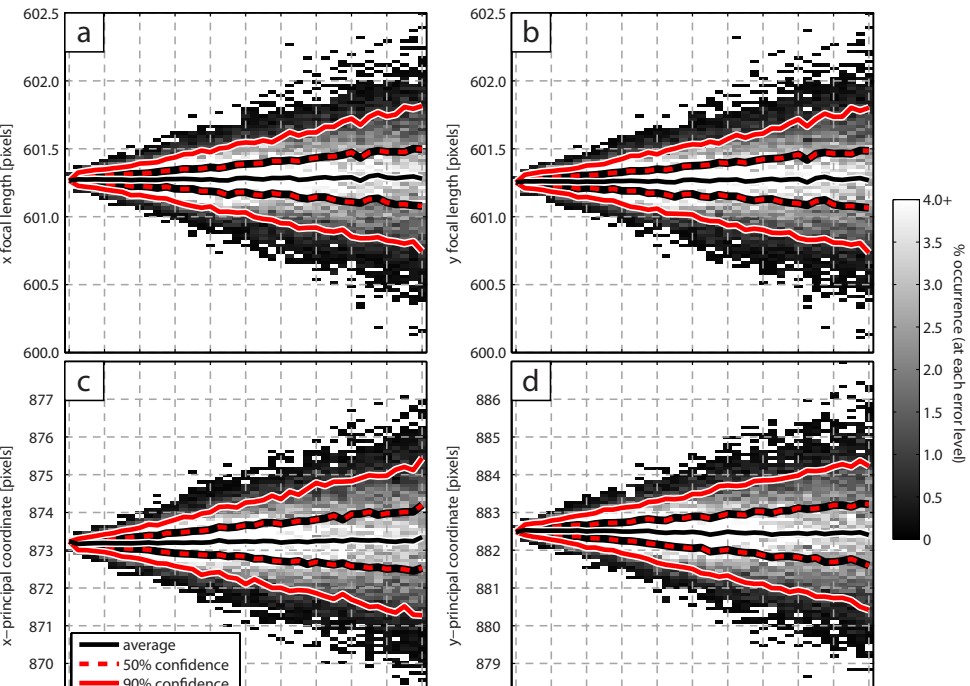

**Figure 5.** Distributions of **(a)** $x$ focal length $\alpha_x$; **(b)** $y$ focal length $\alpha_y$; and **(c)**, **(d)** principal point $(x_o, y_o)$ are shown as a function of measurement error SD$\sigma_j$. The mean, 50 and 90 % confidence intervals are shown as curves.