# Peer review of "Sky camera geometric calibration using solar observations"

_Atmospheric Measurement Techniques, 2015_

## Referee Comment (RC1) · R. Clay (Referee) · 27 Jan 2016

This is an interesting paper which should serve as a useful reference when camera geometries require calibration. This occurs in other areas of research outside of the solar energy industry. The summary of the practical issues relating to image distortion and its subsequent correction are carefully discussed with considerable clarity. The paper presents theoretical background plus an example of the calibration process.

The title is appropriate and the references helpful.

The paper is publishable but I would like to see the incorporation of some thoughts concerning the comments below.

So far as the presentation of this paper is concerned, to the extent that it is in the
[Figure]

context of sky monitoring for the forecast of renewable power generation, I would have liked to be told early on what error levels were required for the system, rather than discovering (page 27) "For the application. . . . . . .these error levels are satisfactory at present.". What would have been unsatisfactory? In other words, do we care that the rms error is around 1 pixel? Is that good enough? Is it overkill?

After, a very interesting and informative first half of the paper, I was left wondering about some things of an implementation nature in section 3 onward.

The practical calibration is based on using the solar disk with its saturated pixels and halo of scattered sunlight. I don't think that we are told whether pixels outside the disk saturate but figure 2 shows large solar-illuminated areas whose symmetry is not discussed. These are presumably due to aerosol scattering and no comment is made about the possibility of selecting both clear and Rayleigh-only days.

It is acknowledged that the solar disk is distorted towards the horizon by atmospheric refraction ('the apparent solar zenith angle must be corrected accordingly' page 16, 'the sun diameter changes with solar zenith angle' page 29) but it appears to be assumed that the center of the 'disk' remains in the appropriate solar direction. This is not likely to be true.

I feel that a fundamental problem with calibrating using the solar path (a natural enough strategy) is that one samples a very limited range of the image plane, which does not quite extend to the zenith outside tropical locations. This may be fine for radially symmetric distortions, but may not be so good for tangential issues. Since the detector saturates on the half degree solar disk, maybe the use of stars or the moon would be useful in filling the image space.

On page 22 line 3, how do you know that you have a global minimum?

There are some ill defined judgement levels which did not feel comfortable: "values that were obviously incorrect" "falsely large skewness" "kn are very close to bn"

---

## Referee Comment (RC2) · Anonymous Referee #2 · 4 Feb 2016

This manuscript describes a procedure for automated geometric calibration of sky imaging cameras. The purpose is to advance localized solar forecasting for solar power generation stations. The manuscript is very well-written and is suitable for publication, but I have a few minor comments that should be addressed first.

General Comments:

1) Days with scattered cloud cover (SCC) appear to be the most important days for the forecasts that motivate this study because high-frequency variability in the downwelling shortwave is comparatively small for both clear-sky and overcast conditions. Thus, it is a bit disappointing that uncertainties in the calibration associated with the sun position measurement during SCC are not more central to the focus of Section 5. What are the errors when there is SCC (as opposed to days with SCC where clear-sky times reduce

the daily mean errors)? One way to communicate this might be to add labels to Figure 4 showing where along the x-axis SCC (or whatever conditions are the most important for the forecasts) likely falls based on the analysis from SGP.

Additionally, it might be worth noting that the uncertainty associated with SCC will increase with solar zenith angle because even if the actual SCC is constant in time and space, there will be an apparent increase in SCC near the horizon from the perspective of the camera (e.g., Warren et al. 1986, p18).

Warren, S.G., C.J. Hahn, J. London, R.M Chervin and R.L. Jenne, (1986),ÂăGlobal Distribution of Total Cloud Cover and Cloud Type Amounts over Land.Âă NCAR Technical Note TN-273+STR, Boulder, CO, 29 pp. + 200 maps

2) None of the errors are translated to Wm-2, which I presume is the metric used for the forecasts and the way that forecast requirements are communicated. The final statement of the text (P27L22-24) suggests you have thought about this some. Can you add some detail to that final statement and also in the introduction that contextualizes the needs of stakeholders? Additionally, the final statement implies that future needs may require more accuracy. I'm surprised by this because the text is discussing errors of tenths of a degree in sun position, yet clouds modulate the downwelling shortwave by 100s of Wm-2 (a substantial fraction of the diurnal cycle).

Specific Comments:

P9L20: Why is N set to 9? Is the result very sensitive to the degree of polynomial used? From a cursory glance, it looks as though Gennery provides little guidance to this choice, but also appears to use a much smaller order polynomial than is used here.

P15L1: Following from the previous comment, more generally N for the back-projection is equal to N from Eq. 9, yes?

P16L10: Is leveling not important?

P16L24: If the Temp/Pres correction is deemed necessary, are the annual averages

sufficient for the correction? Synoptic variability in surface pressure at SGP is about as large as the annual cycle, but both the seasonal and synoptic variability in near-surface air temperature at SGP are quite large. (Obviously, these statements also differ by location.) Also, the uncertainty from refraction should be particularly large for low sun angles.

Section 3.3.: I was at first surprised by the lack of discussion surrounding potential errors in the sun position detection in the images, even in the absence of clouds, such as from variability in aerosols of sub-visible ice clouds. However, I see that some context is provided in Appendix A, in particular in the last paragraph. It might be helpful to work this context also/instead into the text in Section 3.3.

P18L21: Check spelling of Marquardt through the text.

---

## Author Comment (AC1) · 9 Apr 2016

IMPORTANT: Please see PDF supplement for a formatted version of these comments and the figures referenced herein!

======================================

======================================

Final Author Comments

―――――――――――――――――――――――――――

1. Interactive comment, Referee 1 R. Clay (Referee) roger.clay@adelaide.edu.au

[Figure]

**1.1. General Comment 1**

1.1.1. Referee Comment This is an interesting paper which should serve as a useful reference when camera geometries require calibration. This occurs in other areas of research outside of the solar energy industry. The summary of the practical issues relating to image distortion and its subsequent correction are carefully discussed with considerable clarity. The paper presents theoretical background plus an example of the calibration process.

The title is appropriate and the references helpful.

The paper is publishable but I would like to see the incorporation of some thoughts concerning the comments below.

1.1.2. Author Response We thank the reviewer for taking the time to read this (relatively lengthy) article and for the kind words about the work.

1.1.3. Manuscript Changes (no changes)

**1.2. General Comment 2**

1.2.1. Referee Comment So far as the presentation of this paper is concerned, to the extent that it is in the context of sky monitoring for the forecast of renewable power generation, I would have liked to be told early on what error levels were required for the system, rather than discovering (page 27) "For the application:.......these error levels are satisfactory at present."

1.2.2. Author's Response The reviewer is pointing out that we used solar power forecasting as a motivation for the paper, which is indeed the case. Our work is heavily focused on this area, and our need to develop a camera calibration approach suitable for our camera systems was born out of the heavy reliance on geometry in our particular forecast scheme. That being said, while the paper is introduced in the context of solar power forecasting, the goal of this paper was to provide an accurate, online calibration method that could be easily automated for a sensor network after installation

had occurred. We found that the realities of the installation process are that you often only have time to get the system bolted down plugged in, and network connected. Having a calibration approach that can be run on data collected after you leave the system and no one is around is useful, and was the impetus for this work.

The impact of geometric calibration errors on forecast performance depends on cloud size, cloud speed, forecast averaging interval, geometry of cloud and camera relative to the plant etc., and how these quantities are used within the forecasting scheme. As far as we know, a sensitivity analysis for each of the forecast model components has not been performed and thus is not available for us to reference in this case. This is the reason no error levels are provided. Performing a sensitivity analysis specifically for this work would be outside the scope of the calibration scheme presented which is the main focus of the paper.

While we cannot fully address this question, we believe it is an important one. We have added some comments to the paper to briefly summarize the difficulty involved.

1.2.3. Manuscript Changes Added: "Providing detailed and quantitative yet widely applicable specifications for geometric camera calibration in solar energy forecasting applications is difficult. The impact of geometric calibration errors depends on cloud size, cloud speed, forecast averaging interval, geometry of cloud and camera relative to the plant etc. For an illustration of geometrical relationships and sensitivity to cloud height errors see Nguyen and Kleissl (2015)."

1.3. General Comment 3

1.3.1. Referee Comment What would have been unsatisfactory? In other words, do we care that the rms error is around 1 pixel? Is that good enough? Is it overkill?

1.3.2. Author's Response We agree with the reviewer that we should not be the judge on what is satisfactory. We therefore removed all qualifiers and just reported the actual error values. The practitioner would have to judge whether the level of accuracy is

sufficient for a particular application.

1.3.3. Manuscript Changes Removed: "For the application of solar forecasting using sky imagery, these error levels are satisfactory (at present)."

1.4. General Comment 4

1.4.1. Referee Comment After, a very interesting and informative first half of the paper, I was left wondering about some things of an implementation nature in section 3 onward.

1.4.2. Author's Response (no response)

1.4.3. Manuscript Changes (no changes)

1.5. General Comment 5

1.5.1. Referee Comment The practical calibration is based on using the solar disk with its saturated pixels and halo of scattered sunlight. I don't think that we are told whether pixels outside the disk saturate but figure 2 shows large solar-illuminated areas whose symmetry is not discussed. These are presumably due to aerosol scattering and no comment is made about the possibility of selecting both clear and Rayleigh-only days.

1.5.2. Author's Response We agree with the reviewer that the algorithm to detect the sun position is simplistic and this had already been acknowledged in the paper. Our particular camera and the High Dynamic Range (HDR) processing result in a large dynamic range that causes the saturation region to be limited to the solar disk and surrounding halo of scattered sunlight most of the time.

Scattering by elevated aerosol concentrations or thin clouds could cause an expansion of the saturated region. If the expansion brought about asymmetry (as it does when clouds are present) it would negatively affect the accuracy of detecting the sun position. For the case of assymetric saturation in the presence of thin clouds, horizontal homogeneity cannot be assumed and the resulting asymmetry is likely the reason for the fluctuations in sun position on June 7. We hypothesize that aerosol concentrations are typically horizontally homogeneous which together with the axisymmetry of the scattering phase function would maintain a symmetric saturated region. For which the algorithm presented is suitable.

The projection of the camera lens causes radial distortion of the saturated region which is not circular and will cause slight errors in the detection process. In this case, the error in measuring the sun position when assuming a circular shape for the saturated pixels is no more than 2.5 pixels (biased radially inward). This is for the worst case when the region of saturated pixels is large. It is typically much lower, on the order of less than a pixel. This estimate assumes an equisolid angle projection.

1.5.3. Manuscript Changes The beginning of section 3.3 is changed as follows: "Measurement data consists of automated detection of the sun's position xs in an image using a set of methods described in Appendix A. The present method assumes that only pixels in the solar disk and surrounding halo of scattered sunlight saturate and that the saturation region is nearly circular with symmetry about the radial line from the principal point through the sun's center. In practice thin clouds could cause an expansion of the saturation region especially for cameras with low dynamic range and this would reduce the accuracy of the sun position detection. A cloud detection process (e.g. Chow et al. 2011) can be used to discard images with significant cloud cover. The detection process..."

See also changes noted in section 1.6.3.

1.6. General Comment 6

1.6.1. Referee Comment It is acknowledged that the solar disk is distorted towards the horizon by atmospheric refraction ('the apparent solar zenith angle must be corrected accordingly' page 16, 'the sun diameter changes with solar zenith angle' page 29) but it appears to be assumed that the center of the 'disk' remains in the appropriate solar direction. This is not likely to be true.

[Figure]

1.6.2. Author's Response We did indeed assume that the centroid of the distorted solar disk is appropriate to consider as the observed location of the true sun's center (i.e. the refracted location of the sun's center). The concern is of course that refraction causes the bottom half (higher zenith angle) of the sun to be more 'compressed' than the top half (lower zenith angle), which will result in our algorithm measuring the sun centroid at a lower zenith angle than the correct observed location. This does happen and is a limitation of the algorithm. To estimate the magnitude of the effect, we estimated the centroid position of the sun when the sun is nominally supposed to be observed at 89.7° and found that it was lower by 3.6 arcseconds which translates into 0.008 pixels. This was estimated using the refraction model in the NREL solar position algorithm. We don't use images for observed zenith angles greater than 89.625, so 3.6 arcsec should be the upper limit on this error. This error is orders of magnitude smaller than the calibration accuracy and can be safely ignore.

1.6.3. Manuscript Changes Added to first paragraph in Appendix A: "As the sun approaches the horizon, refraction and the lens projection causes a distortion of the solar disk. In both cases, the image of the sun is compressed in the radial direction. When the sun is near the horizon and the sun's pixel are not saturated, refraction causes an estimated 0.008 pixel radial shift of the sun's centroid toward the image center which can safely be ignored. When the saturated sun region is enlarged due to scattering (ocurring when the sun is not near the horizon), the lens distortion of the saturated region can introduce a radial shift of the sun's centroid of up to 2.5 pixels toward the image center. The larger the saturated region, the larger the observed shift, with a typical value near one pixel. These effects appear stable enough to model and correct, however since they are typically on the order of 1 pixel or less, we do not make any adjustments here."

See also changes noted in section 1.5.3.

1.7. General Comment 7

1.7.1. Referee Comment I feel that a fundamental problem with calibrating using the solar path (a natural enough strategy) is that one samples a very limited range of the image plane, which does not quite extend to the zenith outside tropical locations. This may be fine for radially symmetric distortions, but may not be so good for tangential issues. Since the detector saturates on the half degree solar disk, maybe the use of stars or the moon would be useful in filling the image space.

1.7.2. Author's Response Your observation is entirely correct. The input calibration data is quite limited in coverage over the image. As you note, anything outside the tropics will not have data extending to zenith, and terrain may obstruct data near the horizon, leaving you with less than desirable coverage over the image plane. Additionally, the arc shape of the sun path only covers a limited swath of the image plane and the fit of tangential distortion suffers.

Depending on the sensitivity of the camera and the available dynamic range, you may be able to image stars and perform a true stellar calibration with full coverage of the image plane (less terrain obstructions). For cameras that are not able to capture night time images, however, you have to either use a calibration setup, or as we suggest use the sun position. One option is to calibrate the internal and distortion parameters using a checkerboard (e.g. Kannala et al 2006), and use the sun position only for pose. In this case you need only a few measurements of the sun (theoretically, the minimum is 3 noncolinear points), and they don't have to be near zenith or the horizon.

The goal in this work was to provide a "set it and forget" approach and show the resulting accuracy that you can obtain. While it isn't the most accurate calibration method, it is useful if you want to deploy a camera system with minimal effort (see response in section 1.2.2). Using only the sun provides a reasonable means to do this, given the limitations noted.

Kannala, J. and Brandt, S. S.: A generic camera model and calibration method for conventional, wide-angle, and fish-eye lenses, IEEE T. Pattern Anal., 28, 1335–1340,

doi:10.1109/TPAMI.2006.153, 2006.

1.7.3. Manuscript Changes (no change)

1.8. Specific Comment 1

1.8.1. Referee Comment On page 22 line 3, how do you know that you have a global minimum?

1.8.2. Author's Response Finding a for which the projection error is a minimum is the goal, but in practice we do not find the global minimum. The error hypersurface for calibration is very complicated, and from experience appears riddled with local minima that can get the Levenberg-Marquardt stuck. Furthermore, the parameters are not all independent. If they are inversely related in some way, one may go down and the other up with similar calibration error. The best scenario is for a minimum that gives suitable parameter estimates for your model. That was the motivation for using a staged optimization - to walk the optimization through increasing levels of model complexity. This why the basic parameters (i.e. everything except the distortion) were fit first and the local minima obtained by subsequent stages gave basic parameters that were closer to the manufacturer specified values.

1.8.3. Manuscript Changes (no change)

1.9. Specific Comment 2

1.9.1. Referee Comment There are some ill-defined judgment levels which did not feel comfortable: a. "values that were obviously incorrect" b. "falsely large skewness" c. "kn are very close to bn"

1.9.2. Author's Response a. We replaced this statement. b. Skewness was large enough to indicate that pixel axes deviated by over 5 degrees from orthogonal which is not realistic. This was exaggerated in our case because of the coincidental system rotation of about 45 degrees w.r.t. north. We replaced this statement. c. This is an entirely empirical observation that was noticed while doing the work. It was not

thoroughly investigated so we will remove it from the paper. The bn can just as easily be set to zero just as the kn are in the forward calibration process. Statement removed.

1.9.3. Manuscript Changes a. replaced with: "deviated by over 10% from" b. removed: "falsely large" added: "which indicated pixel axes deviated by over 5 degrees from orthogonal" c. original: "To initialize LM for back-projection fitting, coefficients bn can be set to the kn obtained in the forward projection. It was found empirically that kn are very close to bn for the equisolid angle lens used on the USI, and should be even closer if an equidistant lens is used. Coefficients qn can be set to zero."

updated: "To initialize LM for back-projection fitting, coefficients bn and qn can be set to zero."

2. Interactive comment, Referee 1

2.1. General Comment 1

2.1.1. Referee Comment This manuscript describes a procedure for automated geometric calibration of sky imaging cameras. The purpose is to advance localized solar forecasting for solar power generation stations. The manuscript is very well-written and is suitable for publication, but I have a few minor comments that should be addressed first.

2.1.2. Author's Response We appreciate the reviewers complement to the writing.

2.1.3. Manuscript Changes (no change)

2.2. General Comment 2

2.2.1. Referee Comment Days with scattered cloud cover (SCC) appear to be the most important days for the forecasts that motivate this study because high-frequency variability in the downwelling shortwave is comparatively small for both clear-sky and overcast conditions. Thus, it is a bit disappointing that uncertainties in the calibration associated with the sun position measurement during SCC are not more central to

the focus of Section 5. What are the errors when there is SCC (as opposed to days with SCC where clear-sky times reduce the daily mean errors)? One way to communicate this might be to add labels to Figure 4 showing where along the x-axis SCC (or whatever conditions are the most important for the forecasts) likely falls based on the analysis from SGP.

2.2.2. Author's Response We agree that SCC are the most important days for solar forecasts, but the camera calibration is not expected to change from day to day (assumed stationary). Therefore after camera setup, one or a few clear or partially clear days would be used to calibrate the camera parameters and this calibration would be used for all subsequent days. Recalibration would be repeated e.g. quarterly or whenever miscalibration is suspected, e.g. after instrument maintenance.

2.2.3. Manuscript Changes (no change)

2.3. General Comment 3

2.3.1. Referee Comment Additionally, it might be worth noting that the uncertainty associated with SCC will increase with solar zenith angle because even if the actual SCC is constant in time and space, there will be an apparent increase in SCC near the horizon from the perspective of the camera (e.g., Warren et al. 1986, p18).

Warren, S.G., C.J. Hahn, J. London, R.M Chervin and R.L. Jenne, (1986), Global Distribution of Total Cloud Cover and Cloud Type Amounts over Land. NCAR Technical Note TN-273+STR, Boulder, CO, 29 pp. + 200 maps

2.3.2. Author's Response We added a note to that effect.

2.3.3. Manuscript Changes Added: "Due to perspective effects clear periods are less likely to occur for large solar zenith angles (Warren et al. 1986).'

2.4. General Comment 4

2.4.1. Referee Comment None of the errors are translated to Wm-2, which I presume

is the metric used for the forecasts and the way that forecast requirements are communicated. The final statement of the text (P27L22-24) suggests you have thought about this some. Can you add some detail to that final statement and also in the introduction that contextualizes the needs of stakeholders?

2.4.2. Author's Response Due to feedback from the other reviewer, we have removed the statement: "For the application of solar forecasting using sky imagery, these error levels are satisfactory (at present)"

We still believe that the statement is accurate based on experience and intuition, however because we did not perform an analysis on the specific error contributions due to geometric calibration, this statement is more speculation than a documented fact.

Please see our response in section 1.2.2 which also addresses this comment.

2.4.3. Manuscript Changes Please see the changes noted in section 1.2.3.

2.5. General Comment 5

2.5.1. Referee Comment Additionally, the final statement implies that future needs may require more accuracy. I'm surprised by this because the text is discussing errors of tenths of a degree in sun position, yet clouds modulate the downwelling shortwave by 100s of Wm-2 (a substantial fraction of the diurnal cycle).

2.5.2. Author's Response You make a valid point regarding the relative magnitude of the calibration error versus the magnitude of variation in downwelling shortwave. Because of the latter fact, large errors in forecast power production are observed if the cloud condition is not correctly forecast between the sun and the solar collectors. Depending on the size of the power generator (implying the degree of spatial averaging of forecast errors), the impact of the cloud position error from a few pixels of calibration error on the forecast may be quite small when compared to other factors in the forecast. Because the downwelling modulation is so large, if the power generation system is small, a small geometric error can result in large forecast errors when cloud condition

is incorrectly forecast.

For both the reason you bring up here, and the reasons noted in section 1.3.2, the statement has been removed.

2.5.3. Manuscript Changes Please see the changes noted in section 1.3.3.

2.6. Specific Comment 1

2.6.1. Referee Comment P9L20: Why is N set to 9?

2.6.2. Author's Response The order of the polynomial was chosen based on some empirical testing. We reviewed the residuals between the actual radial position of the measurements, and the expected radial position obtained from the nominal equisolid angle projection. As a preliminary step, this assessment required pose, focal length and principal point estimation (using the same procedure outlined in the paper, just with fewer parameters). Upon doing this, we found that a 9th order polynomial gave enough degrees of freedom to match the residual curve over the domain of interest.

2.6.3. Manuscript Changes See next comment 2.7.3.

2.7. Specific Comment 2

2.7.1. Referee Comment Is the result very sensitive to the degree of polynomial used? From a cursory glance, it looks as though Gennery provides little guidance to this choice, but also appears to use a much smaller order polynomial than is used here.

2.7.2. Author's Response Picking an order that is too high will certainly increase the risk of overfitting, and picking an order too low will not always capture the variation of residuals as a function of ray angle from the optical axis. The functional dependence is lens dependent, of course. Order 9 worked for our case and did not appear to have stability issues. Kannala and Brandt (2006) use order 9 as well (odd terms only), however they did so not just for the residuals but for the relation between ray angle and radius in the image plane.

If the expected range of measured ray angles from the optical axis does not provide full coverage of the range , you will almost certainly want to set the polynomial degree to be lower to avoid unwanted behavior in regions where the available data is not constraining the fit. Depending on how (and if) the ray angle is normalized, this may be more or less problematic. We added a statement to address the polynomial order.

Kannala, J. and Brandt, S. S.: A generic camera model and calibration method for conventional, wide-angle, and fish-eye lenses, IEEE T. Pattern Anal., 28, 1335–1340, doi:10.1109/TPAMI.2006.153, 2006.

2.7.3. Manuscript Changes Original: "In this work, was set to nine."

Updated: "In this work, was set to nine. The choice was made based on a review of the residual radial distortion, and was appropriate for the lens examined. A lower order polynomial should be considered if calibration input data does not provide full zenith angle coverage (i.e. ). For more stable curve fitting results, should be appropriately normalized."

2.8. Specific Comment 3

2.8.1. Referee Comment P15L1: Following from the previous comment, more generally N for the back-projection is equal to N from Eq. 9, yes?

2.8.2. Author's Response Yes, in our work it is (P15L01). The order of the polynomial is subject to the same considerations noted above.

2.8.3. Manuscript Changes (no change)

2.9. Specific Comment 4

2.9.1. Referee Comment P16L10: Is leveling not important?

2.9.2. Author's Response Purely for calibration, leveling is not important because as part of the calibration process, the solar position input in the world coordinate frame is rotated into the camera coordinate frame where we can take advantage of distortion

symmetry for the camera model. However, not being level does have disadvantages. If the camera leveling was especially poor, say the optical axis was pointed 10 degrees off from zenith (this is obviously quite extreme for a typical sky camera), you would likely not capture the entire sky with a 180 deg. field of view lens, which may be an issue depending upon the application. If this were the case, it may prevent image capture of the sun at certain angles where you might be able to image it if the camera were level, reducing available data for calibration. The other concern is that distortion is primarily radial about the optical axis, so if the optical axis does not coincide closely with zenith, there will be different amounts of distortion as a function of azimuth for a given zenith angle. This may be problematic depending on what you intend to use the imagery for. Purely from a camera calibration perspective, aside from missing sun data, the object is to calibrate the camera in any pose, and the method given will work, but for sky imagery research and other applications it is definitely desirable to level the instrument.

2.9.3. Manuscript Changes (no changes)

2.10. Specific Comment 5

2.10.1. Referee Comment P16L24: If the Temp/Pres correction is deemed necessary, are the annual averages sufficient for the correction? Synoptic variability in surface pressure at SGP is about as large as the annual cycle, but both the seasonal and synoptic variability in near surface air temperature at SGP are quite large. (Obviously, these statements also differ by location.) Also, the uncertainty from refraction should be particularly large for low sun angles.

2.10.2. Author's Response To address this concern, we evaluated the refraction model used in the NREL solar position algorithm, and compared it to the work of other authors. Up to 80 degrees zenith angle, the model follows other models relatively well (Figure R1) given the relative simplicity of the model.

Figure R1 Deviation of various refraction models from the Star Almanac (as reported

in Hohenkerk and Sinclair 1985).

Beyond 80 degrees zenith angle, the deviations are still reasonable (Figure R2). The two Hohenkerk models presented in Figure R2 are for two different lapse rates: K/m (Hohenkerk 1) and (Hohenkerk 2), which gives an idea of the impact of lapse rate on refraction. NREL's refraction model deviation from the reference selected here is similar to that caused by the temperature gradient.

(a) (b) Figure R2 Deviation of various refraction models from Garfinkel's model (as reported in Hohenkerk and Sinclair 1985) for units of (a) degrees and (b) pixels for our camera. Negative indicates that the refraction correction is less than that prescribed in the reference model. In the case of the NREL model, negative indicates the reportedly observed zenith angle higher than is actually observed (i.e. it is biased towards the horizon).

Because the modeling error is well below a pixel, variations due to temperature and pressure are not as significant in this application as they might be in others. The order of the refraction error from atmospheric conditions is much less than the error introduced by the sun detection process implemented for the paper.

We will adjusted our mention of refraction slightly to illuminate the effects of changing atmospheric parameters.

2.10.3. Manuscript Changes original: "The uncertainty on solar zenith angle reported by Reda and Andreas is $\pm 0.0003°$, and if the image capture time is one second off, the error in solar hour angle would be $\pm 0.004°$. In comparison, for our lens a one pixel measurement uncertainty in sun position measurements corresponds to approximately $= 0.14°$ at the horizon."

updated: "Using the nominal pressure and temperature values is a suitable choice since a change in lapse rate from 0.0057 K/m to 0.0065 K/m can introduce a difference in refraction of only $= 0.007°$ at $= 90°$ (Hohenkerk and Sinclair, 1985). (Considering

this, it should be noted that the $\pm 0.0003°$ uncertainty on solar zenith angle reported by Reda and Andreas is quite generous.) A one second error in image capture time results in an error of $\pm 0.004°$ for solar hour angle. In comparison, for our lens a one pixel uncertainty in sun position measurements corresponds to approximately $= 0.14°$ at the horizon."

2.11. Specific Comment 6

2.11.1. Referee Comment Section 3.3.: I was at first surprised by the lack of discussion surrounding potential errors in the sun position detection in the images, even in the absence of clouds, such as from variability in aerosols of sub-visible ice clouds. However, I see that some context is provided in Appendix A, in particular in the last paragraph. It might be helpful to work this context also/instead into the text in Section 3.3.

2.11.2. Author's Response We have added some additional context to section 3.3 (see also, section 1.5 of this document).

2.11.3. Manuscript Changes The beginning of section 3.3 is changed as follows: "Measurement data consists of automated detection of the sun's position xs in an image using a set of methods described in Appendix A. The present method assumes that only pixels in the solar disk and surrounding halo of scattered sunlight saturate and that the saturation region is symmetric about the radial line from the principal point through the sun's center. In practice thin clouds could cause an expansion of the saturation region especially for cameras with low dynamic range and this would reduce the accuracy of the sun position detection. The detection process..."

2.12. Specific Comment 7

2.12.1. Referee Comment P18L21: Check spelling of Marquardt through the text.

2.12.2. Author's Response Noted.

2.12.3. Manuscript Changes Spelling corrected.

Please also note the supplement to this comment:
http://www.atmos-meas-tech-discuss.net/amt-2015-277/amt-2015-277-AC1-supplement.pdf
[Figure]

**Fig. 1.**

[Figure]

**Fig. 2.**